# AGREE TO DISAGREE: DEMYSTIFYING HOMOGENEOUS DEEP ENSEMBLE THROUGH DISTRIBUTIONAL EQUIVALENCE

**Yipei Wang, Xiaoqian Wang** [*]
Elmore Family School of Electrical and Computer Engineering
Purdue University
West Lafayette, IN 47907, USA
`{wang4865,joywang}@purdue.edu`

## ABSTRACT

Deep ensembles improve the performance of the models by taking the average predictions of a group of ensemble members. However, the origin of these capabilities remains a mystery and deep ensembles are used as a reliable "black box" to improve the performance. Existing studies typically attribute such improvement to Jensen gaps of the deep ensemble method, where the loss of the mean does not exceed the mean of the loss for any convex loss metric. In this work, we demonstrate that Jensen's inequality is **not** responsible for the effectiveness of deep ensembles, and convexity is not a necessary condition. Instead, Jensen Gap focuses on the "average loss" of individual models, which provides no practical meaning. Thus it fails to explain the core phenomena of deep ensembles such as their superiority to any single ensemble member, the decreasing loss with the number of ensemble members, etc. Regarding this mystery, we provide theoretical analysis and comprehensive empirical results from a statistical perspective that reveal the true mechanism of deep ensembles. Our results highlight that deep ensembles originate from the homogeneous output distribution across all ensemble members. Specifically, the predictions of homogeneous models (Abe et al., 2022b) have the **distributional equivalence** property – Although the predictions of independent ensemble members are point-wise different, they form an identical distribution. Such agreement and disagreement contribute to deep ensembles' "magical power". Based on this discovery, we provide rigorous proof of the effectiveness of deep ensembles and analytically quantify the extent to which ensembles improve performance. The derivations not only theoretically quantify the effectiveness of deep ensembles for the first time, but also enable estimation schemes that foresee the performance of ensembles with different capacities. Furthermore, different from existing studies, our results also point out that deep ensembles work in a different mechanism from model scaling a single model, even though significant correlations between them have been observed. The code to re-implement all the experiments is open source at https://github.com/yipei-wang/DeepEnsembleDemystified.

## 1 INTRODUCTION

In the studies of deep learning models, pursuing high performance is always one of the ultimate goals, where an enormous body of approaches regarding model architecture, training regimes or data augmentations, etc. have been proposed (Krogh & Hertz, 1991; Dietterich, 2000; Mnih et al., 2013; Simonyan & Zisserman, 2014; Kingma & Ba, 2014; Lee et al., 2015; He et al., 2016; Vaswani et al., 2017; Tolstikhin et al., 2021). Among these approaches, ensembling a group of models has been demonstrated very effective in improving the performance of models (Schapire, 1990; Breiman, 1996; 2001; Chen & Guestrin, 2016; Ganaie et al., 2022). Different from traditional boosting/bagging methods, *deep ensembles*, on the other hand, form the ensemble of models where the differences between ensemble members are only caused by the randomness of the stochastic gradient descent (SGD) optimizations, i.e., the random initialization of models and the order of the presence of training

---
[*]corresponding author

data. (Lee et al., 2015; Huang et al., 2017; Lakshminarayanan et al., 2017). This approach is not only simple but also very effective in improving the performance of models (Fort et al., 2019; Kondratyuk et al., 2020) and the robustness (Chen et al., 2023; Huang et al., 2023). It also sheds light on various applications such as bioinformatics (Cao et al., 2020), geoscience (Zeng et al., 2023), etc.

However, effective and powerful as it is, the source of the capability of deep ensembles still remains a mystery. Many existing works thereby aim at exploring the true mechanisms of the deep ensemble method. Some studies focus on the mechanisms of deep ensembles via the relations among ensemble members, including their diversities, agreements, etc. (Fort et al., 2019; Theisen et al., 2023). Mathematical modeling for the relation between the calibrated negative log-likelihood (NLL) and the number of ensemble members is also proposed (Lobacheva et al., 2020). Discouragingly, in recent work, it is discovered that the ensemble diversity is neither responsible for improved uncertainty quantification nor the improved robustness (Abe et al., 2022b). Their findings conclude that deep ensembles still remain a reliable "black-box" method to improve the performance of DNNs. Whether the performance of large single models and an ensemble of smaller models originate from the same source remains unknown, too (Lakshminarayanan et al., 2017; Fort et al., 2019; Lobacheva et al., 2020; Kondratyuk et al., 2020; Abe et al., 2022a;b).

Different from existing work, we study this problem from a statistical, and global perspective. **Model Distribution:** Because of the over-parameter nature, modern DNNs are capable of overfitting the training set (Goodfellow et al., 2014; Allen-Zhu et al., 2019; Nakkiran et al., 2021). Given a fixed model structure, stochastically trained models form specific distributions over the functional space. Training single models is then equivalent to uniformly drawing samples (i.e. trained models) from this distribution (Lakshminarayanan et al., 2017; Gawlikowski et al., 2023). As a consequence, deep ensembles, by taking the mean of the predictions of all $M$ ensemble members, are implicitly estimating the expectation over such distributions. Different models make different predictions given the same samples, resulting in performance improvement in their means. **Data Distribution:** On the other hand, predictions for different samples also differ for a given model, forming a distribution of predictions over the output space. Existing work focuses on the diversity brought by the models, but overlooks the importance of the distribution caused by the stochasticity in the data distribution. The resulting point-wise evaluation of the model diversity $\text{Var}_F[F(\boldsymbol{x})]$ fails to reveal the true mechanism of deep ensembles. We consider the stochasticity from the joint distribution of both models and data and how the output distributions are affected. Through this approach, we provide a rigorous and theoretical analysis of deep ensembles. Beyond the derivations, we point out that existing studies suffer from insufficient sample size (i.e. the number of ensemble members) in studying the statistics of high-dimensional and complex distributions. To resolve this issue and provide a comprehensive empirical validation of theoretical results, we study deep ensembles via $M = 100$ ensemble members across various datasets and model structures. We also incorporate the long-standing debate between the effectiveness of deep ensembles and large single models (Lakshminarayanan et al., 2017; Fort et al., 2019; Lobacheva et al., 2020; Kondratyuk et al., 2020; Abe et al., 2022a;b).

Previous work has attributed the improvement in deep ensembles to the Jensen gap (e.g. (Abe et al., 2022b)), which is the difference between the loss of the ensemble (average) of individual models and the average loss of all individual models. When the loss metric is convex (e.g. Brier score, negative log-likelihood, etc.), this value is positive according to Jensen's inequality, which means that the loss of the ensemble is guaranteed to be smaller than the "average loss". However, we argue that *Jensen's inequality is insufficient in explaining the predictive improvement of deep ensembles*. This is because the average loss of a collection of models has no practical meaning — what makes deep ensembles effective is the superiority to the loss of *any* single model instead of the average loss. And this is not affected by the Jensen gap. Besides, it is also consistently observed that increasing the number of ensemble members improves the performance, which is also irrelevant to the Jensen gap. In this work, we uncover a property of trained models in deep ensembles called **distributional equivalence**. That is, the homogeneous ensemble members, although differ in the predictions of every sample, form the same distributions over the entire data distribution. We thus provide both theoretical analysis and comprehensive empirical results to demonstrate that it is this distributional equivalence across ensemble members that contributes to *all the observed behaviors* of deep ensembles. The paper is organized as follows. In section 2, we first review the recent efforts in uncovering the mechanism of deep ensembles and summarize their differences with our work. Then we introduce the setups and summarize the deficiency of existing understanding of deep ensembles in section 3. Afterward, we introduce the main theoretical findings of distributional equivalence in sections 4 and 5 and delve

into the essence of deep ensembles. For the compactness, all proofs are deferred to appendix B. The main contributions of this work can be summarized as follows:

- We focus on understanding the mechanism of deep ensembles, and point out the problems of the existing approach that is based on Jensen's inequality.
- We reveal the distributional equivalence property of trained models and provide a theoretically proven explanation for the effectiveness of deep ensembles.
- Our theoretical findings provide schemes to accurately estimate the ensemble performance regarding the number of ensemble members and to use only two models to foresee the asymptotic loss of deep ensembles with infinite capacity.
- All theoretical findings are verified through extensive experiments carried out in this work.

## 2 RELATED WORK

Intrigued by the simplicity of implementations and the impressive performance, many studies have been carried out focusing on the internal mechanism of deep ensembles and trying to explore the reason why it works. We hereby summarize these most related works and how our approach and discovery are connected to and different from them.

Lobacheva et al. (2020) focus on the pattern of the negative log-likelihood (NLL) of deep ensembles in classification tasks regarding the number of ensemble members. An empirically verified power law is proposed for the calibrated NLL. The power law provides a guideline for designing the ensemble system and determining the most efficient number of ensemble members. However, one drawback of this work is that it's fully empirical, which can give rise to trustworthiness issues in the results. Abe et al. (2022a;b) provide insightful studies of deep ensembles on the improved uncertainty quantification (UQ) and the improved robustness. However, it is concluded that the Jensen gap measures the expected predictive improvement through ensembling, which we demonstrate in the later section is insufficient. Besides, Abe et al. (2022b) also admits that deep ensemble is still a "reliable 'black box'" to improve the performance of DNNs. Another similar but slightly different ensembling scheme is through majority vote (Masegosa et al., 2020; Theisen et al., 2023), where the final prediction is obtained by the vote of all ensemble members instead of the average of them. However, the theoretical studies for this variant are limited by its generality — when all the ensemble members are considered general functions, the conclusions become very weak. For example, Theisen et al. (2023) point out that the theoretical error of the majority vote classifier can be even worse than the average error rate, which is far from the empirical observations. This indicates that important conditions satisfied by the distribution of the ensemble members and the data are overlooked, which we highlight in this work. As for the empirical study, the majority vote typically requires training a larger number of ensemble members compared with standard deep ensembles.

Many existing works also point out the similarity between performance improvement from (1) increasing the capacity of a single model and (2) ensembling many small models (e.g. (Lakshminarayanan et al., 2017; Abe et al., 2022b), etc.), where empirical comparisons between them are carried out. Geiger et al. (2020); Kobayashi et al. (2021) carry out empirical studies and discover a specific bias-variance trade-off curve for deep ensembles, where increasing the number of ensemble members after the optimal ensemble underscores the ensemble performance. This phenomenon is also explained by the distributional equivalence property highlighted in our work.

## 3 DEFICIENCIES IN EXISTING STUDIES

### 3.1 SETUPS

**Notations.** In this work, we focus on the classification problem with the dataset $\mathcal{D} = \mathcal{X} \times \mathcal{Y}$, where $\mathcal{X} \subset \mathbb{R}^d$ and $\mathcal{Y} = \{1, \cdots, c\}$. Here $d$, $c$ denote the input dimension and the number of classes, respectively. Let $\boldsymbol{f} : \mathbb{R}^d \to \Delta^{c-1}$ denote the model that maps from the input space to the standard $(c-1)$-simplex. Such models are optimized stochastically with the objective $\min_\theta \mathbb{E}_{(\boldsymbol{x},y) \in \mathcal{X}_{\text{train}} \times \mathcal{Y}_{\text{train}}} L(\boldsymbol{f}_\theta(\boldsymbol{x}), y)$, where $\theta$ denotes the weights of $f$ and is omitted for simplification. Specifically, let $f_y : \mathbb{R}^d \to [0, 1]$ denote the predicted probability of the target class, where the subscript $y$ is omitted in notations for compactness. Based on the model $f$, the metrics are defined as $\phi : (0, 1) \to \mathbb{R}_+$, which is determined by the specific type of metrics. For example, the Brier score is defined as $\phi_B(f(\boldsymbol{x})) = (1 - f(\boldsymbol{x}))^2$ [1]. And for negative log-likelihood (NLL),

---

[1] The original Brier score is defined over the entire prediction as $\|\boldsymbol{f}(\boldsymbol{x}) - \mathbf{1}_y\|_2^2$. Here we focus on the attribute to the target class. The differences are discussed in appendix B.10

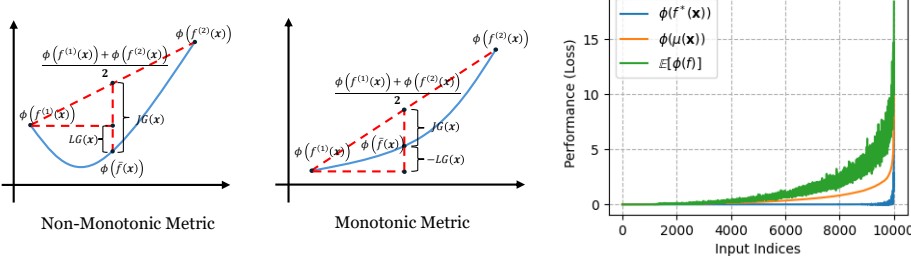

(a) Non-Monotonic Metric vs Monotonic Metric.

(b) Point-Wise LG vs JG

Figure 1: (a) An illustration of $JG$ vs $LG$ in the $M = 2$ scenario. Since $JG(\boldsymbol{x}) \geq 0$ always holds, it provides no insights into how deep ensembles work. On the other hand, $LG(\boldsymbol{x}) \leq 0$ is possible only for non-monotonic metrics. (b) The point-wise prediction of the best single model (blue), ensemble (orange), and the average error (green). Samples are sorted by the ensemble predictions. The results are generated from the CNN model with $k = 10$ and CIFAR-10.

$\phi_{nll}(f(\boldsymbol{x})) = -\log f(\boldsymbol{x})$. The models are evaluate on $\mathcal{X}_{\text{test}}, \mathcal{Y}_{\text{test}}$ throughout the paper. Thus for brevity, we omit the subscripts.

**Homogeneous Ensembles.** The deep ensemble approach is implemented by taking the average of a group of independently optimized models as the final predictions. Homogeneous ensembles further refer to the variant where all ensemble members have identical structures and training hyper-parameters and only differ in the training stochasticity. Therefore, a trained ensemble member $F \sim p_F$ follows a distribution $p_F$, which is a probability measure over the functional space $\mathcal{F}$. And $\text{supp}(p_F) \subseteq \mathcal{F}$ denotes the support of $p_F$. Formally, homogeneous ensembles take a set of $M$ models $\{F = f^{(i)}\}_{i=1}^{M}$, which are considered i.i.d. in $p_F$, and predict in the following way:

$$\bar{f}(\boldsymbol{x}) = \mathbb{E}_{F \sim p_F}[F(\boldsymbol{x})]; \quad \bar{f}(\boldsymbol{x}; M) = \frac{1}{M} \sum_{i=1}^{M} f^{(i)}(\boldsymbol{x}). \tag{1}$$

Surprisingly, such a simple scheme can drastically improve the performance (i.e., decrease the loss value). Since all members of the ensemble share an identical structure, the optimization process can be viewed as random sampling from the model distribution $p_F$.

**Experiment Setups.** To empirically validate our theoretical findings, we carry out comprehensive experiments, which are presented along with theoretical results. Each model structure determines a model family $\mathcal{F}$ and the distribution $p_F$ over $\mathcal{F}$. Varying the model architectures in CNNs and ResNets and capacities in width determined by $k \in \{10, 20, 40, 80, 160\}$, we include a total of 10 $p_F$ in our experiments. For each $p_F$, we train $M = 100$ models for three datasets: CIFAR-10/100 and TinyImagenet. i.e., a total of $2 \times 5 \times 3 \times M = 3000$ trained models. The training parameters follow the suggestions in Nakkiran et al. (2021). The setup for experiments is detailed in appendix A.

### 3.2 Understanding Deep Ensembles through Jensen's Inequality

Existing analysis regarding deep ensembles attributes the improvement to the Jensen Gap. Let $\boldsymbol{x} \in \mathcal{X}$ be an input data. Then given the functional distribution $p_F$ of the homogeneous models and a convex metric $\phi$, it is according to Jensen's inequality that

$$\phi\big(\mathbb{E}_{F \sim p_F}[F(\boldsymbol{x})]\big) \leq \mathbb{E}_{F \sim p_F}\big[\phi(F(\boldsymbol{x}))\big] \tag{2}$$

That is, for any single input $X \sim p_X$, the "error" of the ensemble is no greater than the average error among all the members of the ensemble. Then the Jensen gap $JG(\boldsymbol{x}) = \mathbb{E}_{F \sim p_F}[\phi(F(\boldsymbol{x}))] - \phi(\mathbb{E}_{F \sim p_F}[F(\boldsymbol{x})]) \geq 0$ is used as a measurement for the "predictive improvement" of the ensemble scheme. For example, Abe et al. (2022b) point out that for Brier score $\phi_B$, the Jensen gap is equivalent to the point-wise variance of the model distribution $p_F$ at $X = \boldsymbol{x}$.

### 3.3 The Failure of Evaluating the "Average Loss"

While eq. (2) provides a *seemingly* reasonable and plausible explanation for the effectiveness of the deep ensemble scheme, it fails in practice. An important deficiency is that the average loss $\mathbb{E}_{F \sim p_F}[\phi(F(\boldsymbol{x}))]$ does not possess any practical meaning, and thus having lower loss values than

Table 1: A demonstration of the three explanations for the effectiveness of deep ensembles. The point-wise Jensen gap always holds, but has no practical meaning. The point-wise lower gap being positive can be strong evidence for the effectiveness of deep ensembles. However, it never holds. The global gap defines why deep ensembles are always preferred. And it is proved in this work.

| Types | Formula | Positivity | Significance | Feasibility |
|---|---|---|---|---|
| Jensen Gap | $\mathbb{E}_{F\sim p_F}[\phi(F(\boldsymbol{x}))] - \phi(\mathbb{E}_{F\sim p_F}[F(\boldsymbol{x})])$ | Always $\geq 0$ | $\times$ | $\checkmark$ |
| Lower Gap | $\min_{f\in\mathcal{F}}\{\phi(f(\boldsymbol{x})\} - \phi(\mathbb{E}_{F\sim p_F}[F(\boldsymbol{x})])$ | Always $\leq 0$ | $\checkmark$ | $\times$ |
| Global Gap | $\min_{f\in\mathcal{F}}\{\mathbb{E}_X[\phi(f(\boldsymbol{x}))]\} - \mathbb{E}_X[\phi(\mathbb{E}_F[F(X)])]$ | Proved to be $\geq 0$ | $\checkmark$ | $\checkmark$ |

this mean does not contribute to the success of deep ensembles. Jensen's inequality always holds, regardless of the instantiations of the ensemble members. Thus using Jensen Gap to explain deep ensembles' effectiveness suggests that *for any group of models* $\{f^{(i)}\}_{i=1}^M$, *the ensemble of them is always desired.* This is trivially false. On the contrary, Jensen's inequality eq. (2) allows the possibility that $\exists f^* \in \mathcal{F}$ s.t. $\phi(f^*(\boldsymbol{x})) < \phi(\mathbb{E}_{F\sim p_F}[F(\boldsymbol{x})])$. In such scenarios, some single ensemble members may outperform the ensemble itself. Note that an ensemble is useful **only if it outperforms all ensemble members**. Therefore, in practice, what really matters is the difference between the best individual (minimized loss) and the ensemble, which is denoted as the lower gap (LG):

$$LG(\boldsymbol{x}) = \min_{f\in\mathcal{F}}\big\{\phi(f(\boldsymbol{x}))\big\} - \phi\big(\mathbb{E}_{F\sim p_F}[F(\boldsymbol{x})]\big) \tag{3}$$

LG provides a more reasonable intuition for the ensemble. It suggests that deep ensembles are desired if they outperform **all** individuals. This provides practical meanings compared with JG.

### 3.4 THE FAILURE OF POINT-WISE EVALUATION

**Point-Wise Gap.** Note that although $LG$ provides a more practical understanding compared with $JG$, $LG(\boldsymbol{x}) \geq 0$ can be an exceptionally strong condition to satisfy. In fact, $LG(\boldsymbol{x}) \geq 0$ is possible only if the metric $\phi(\cdot)$ is **not** monotonic. Formally, we prove the following proposition in appendix B.1.

**Proposition 3.1.** $LG(\boldsymbol{x}) \leq 0$ *always holds for any monotonic metric* $\phi$. *That is, given input sample* $\boldsymbol{x}$, *deep ensemble always underperforms the best individual member.*

The $M = 2$ scenario is visualized in fig. 1 (a). It can be observed that the monotonicity of the metric $\phi$ prohibits $LG(\boldsymbol{x})$ from being positive since the average prediction is always no greater than the highest prediction (i.e. higher loss). fig. 1(b) shows an empirical result of the NLLs of CIFAR-10 and CNN models. The 10000 testing samples are sorted by the ensemble prediction for better presentations. It verifies that $LG(\boldsymbol{x}) \leq 0$ and $JG(\boldsymbol{x}) \geq 0$ always hold.

**Global Evaluation.** Although deep ensembles always underperform at least one individual member for any input sample, the ensemble scheme is still plausible in practice. This shows the deficiency of the point-wise evaluation. What is always overlooked in the studies of deep ensembles is the importance of the distributions of the data $p_X$, whose support is $\mathcal{X}$. Ensemble is desired if the **Global Gap** (GG) is always positive:

$$GG = \min_{f\in\mathcal{F}}\big\{\mathbb{E}_{X\sim p_X}[\phi(f(\boldsymbol{x}))]\big\} - \mathbb{E}_{X\sim p_X}\big[\phi(\mathbb{E}_{F\sim p_F}[F(X)])\big] \geq 0 \tag{4}$$

However, universally one can only prove that both RHS and LHS are less or equal to $\mathbb{E}_{X\sim p_X}\big[\mathbb{E}_{f\sim p_F}[\phi(f(\boldsymbol{x}))]\big]$, which means eq. (4) does not hold in the universal sense. Instead, *it is the distribution of* $(X, F) \sim p_X \otimes p_F$ *that determines whether deep ensembles are beneficial.* The three types of explanations are compared more straightforwardly in table 1.

## 4 DEMYSTIFYING DEEP ENSEMBLES

### 4.1 THE DISTRIBUTIONAL EQUIVALENCE PROPERTY

We start the investigation of $p_X \otimes p_F$ by introducing the **distributional equivalence** property, where the prediction forms identical distributions. Since $F \sim p_F, X \sim p_X$, the predicted probability $\ell = F(X)$ is also a random variable $\ell \sim p_\ell$. Distributional equivalence refers to the property that for any two models $f^{(1)}, f^{(2)} \in \mathcal{F}$, the conditional variable $\ell|F = f^{(1)}$ and $\ell|F = f^{(2)}$ follow the **same** distribution. In fig. 2(a), the kernel density estimation of $\ell|F = f^{(i)}$ for all 10000 samples of CIFAR-10 are visualized. There are 100 individual models, distinguished by the colors of curves. It can be qualitatively observed that $P(\ell|F = f^{(i)}) \approx P(\ell|F = f^{(j)})$ for any $f^{(i)}, f^{(j)} \in \mathcal{F}$.

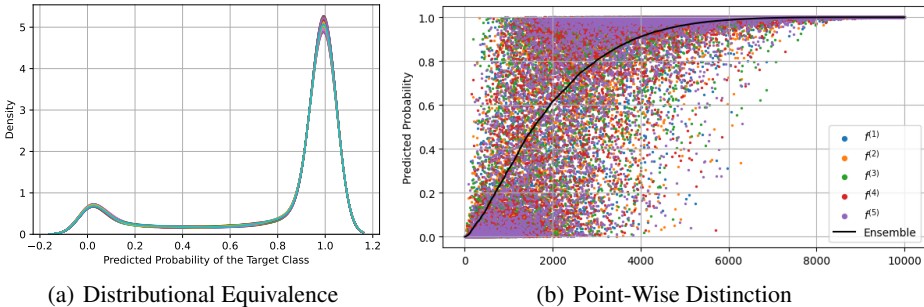

(a) Distributional Equivalence         (b) Point-Wise Distinction

Figure 2: The demonstration of (a) the distributional equivalence property and (b) the point-wise distinction property. The kernel density estimations $\ell|F = f^{(i)}, i = 1, \cdots, M$ for $M = 100$ models are plotted in (a). Conditioning on different models results in the identical distribution of predictions. The point-wise predictions are presented in (b), where 10000 testing samples are arranged according to the prediction of the ensemble (black curve). The point-wise predictions of the first 5 individual models are plotted in different colors. Each point represents a specific prediction $f^{(i)}(\boldsymbol{x}_j)$. It is demonstrated clearly that although models tend to agree on certain samples, the point-wise predictions vary significantly. The models are all CNN with $k = 40$, and the dataset is CIFAR-10.

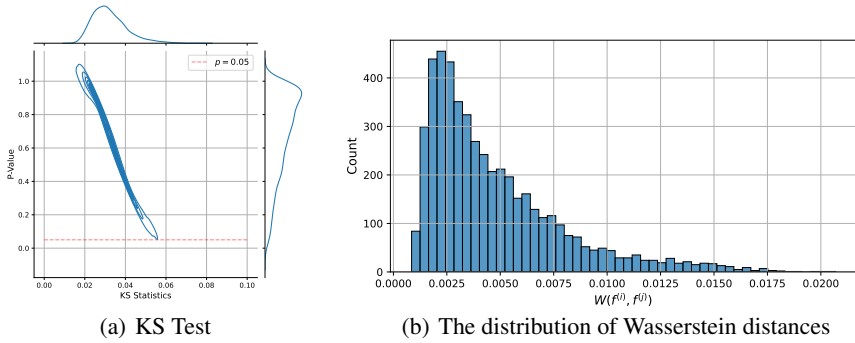

(a) KS Test         (b) The distribution of Wasserstein distances

Figure 3: The quantitative results in validating the distributional equivalence property. $\binom{100}{2} = 4950$ model pairs are evaluated. (a) shows the joint distribution of the $p$-values and KS statistics, where the red dashed line shows the level of $p = 0.05$. (b) shows the distribution of the 4950 Wasserstein distances between model pairs.

We employ two metrics to validate the distributional property of $\ell|F = f$. The Kolmogorov-Smirnov test evaluates the null hypothesis that the conditional distributions $\ell|F = f^{(1)}$ and $\ell|F = f^{(2)}$ are drawn from the same underlying distribution. While the KS test is powerful, it's important to note that the null hypothesis becomes overly strong with the increases in sample size, potentially leading to statistically significant results for practically insignificant differences in large datasets. To complement this, we carry out the KS test over the first 1000 testing samples and also utilize the Wasserstein distance to quantify the minimal cost of transforming one probability distribution into another. This metric provides a continuous measure of dissimilarity between distributions, offering a more nuanced comparison of the conditional distributions than the binary outcome of the KS test. The results are presented in fig. 3. Based on $M = 100$ models sampled from $p_F$, there are $\binom{100}{2} = 4950$ pairs in KS tests and Wasserstein distances. It is observed from (a) that the statistics are all extremely small, with almost all $p$-values greater than 0.05. From (b), the Wasserstein distances are very small compared with the domain of probability $[0, 1]$. Both (a)(b) verifies the assumption that $P(\ell|F = f^{(i)}) \approx P(\ell|F = f^{(j)})$ for any $f^{(i)}, f^{(j)} \in \mathcal{F}$. These results demonstrate that within a fixed model distribution $p_F$ (i.e. fixed training data and stochastic training schemes), the trained models have the same predicted distribution $\ell|F = f$. This is weaker than point-wise equivalence, but much stronger than the equivalence under metrics (e.g. NLLs, Brier scores, etc.).

**Point-Wise Distinction.** Note that distributional equivalence does not imply point-wise equivalence. On the contrary, the point-wise diversity $\text{Var}_{F \sim p_F}(F(\boldsymbol{x}))$ among ensemble members is always considered an important factor (Fort et al., 2019; Abe et al., 2022a; Theisen et al., 2023). As the

point-wise diversity approaches zero, all the ensemble members become point-wise identical and thus the ensemble becomes equivalent to each model. Therefore, it is the point-wise distinction and the distributional equivalence together that contribute to ensembles' power. The point-wise predictions are visualized in fig. 2(b). The 10000 testing samples are sorted by the predicted probability of the target class, where the black curve represents the ensemble prediction. The scatter points represent the predictions of 5 individual ensemble members. It is observed that different models have distinct point-wise predictions but share similar trends. More figures are shown in appendix D.

## 4.2 THE DISTRIBUTION OF MODELS

Although point-wise diversity plays an important role in ensembles, being fixated on a single $x \in \mathcal{X}$ fails to explore the true mechanism of ensembles. This is because globally, the distributions of $\ell = f(X)$ are always affected by the neural collapse phenomenon. According to the overfitting property of modern deep models, a trained model $f \sim p_F$ achieves 100% accuracy on the training set and the training loss becomes sufficiently small. The studies of Neural Collapse (NC) further demonstrate that in the terminal phase of training, discriminative models tend to collapse to the exact dimension of the target space (Papyan et al., 2020). For classification tasks, the predicted probability collapses and converges to either zeros or ones as the model becomes overconfident in both correct and wrong predictions (Hein et al., 2019).

As a result, we consider the trained model distribution $p_F$ under complete neural collapse. That is, $\hat{\mathcal{F}} = \{\hat{f}(x) = \mathbb{1}_{\{\arg\max f(x)=y\}} | f \in \mathcal{F}\}$. The corresponding distribution defined on $\hat{\mathcal{F}}$ is written as $\hat{p}_F(F = \hat{f}) = p_F(\{f \in \mathcal{F} | \mathbb{1}_{\{\arg\max f(\mathcal{X})=\mathcal{Y}\}} = \hat{f}\})$. This can be seen as an approximation of the limit of $p_F$ as the training procedure continues infinitely. Due to the infeasibility of the continuous distribution of $\ell = F(X)$, we utilize $\hat{p}_F$ as a surrogate to $p_F$ in some theoretical derivations. It should be noted that this does not sacrifice the practicality of the theoretical analysis because of the high resemblance between $p_F$ and $\hat{p}_F$. In fact, **all** empirical results are carried out using $p_F$, demonstrating the transferability between $p_F$ and $\hat{p}_F$. The ensemble under the complete neural collapse is defined as

$$\bar{\hat{f}}(x) = \mathbb{E}_{F \sim \hat{p}_F}[F(x)] = \mathbb{E}_{F \sim p_F}[\mathbb{1}_{\arg\max F(x)=y}] \tag{5}$$

Thus for the collapsed approximation $\hat{p}_F$ of models, the conditional distributions of the prediction under fixed sample $x$ or fixed model $\hat{f}$ can collapse to Bernoulli distributions: $\ell | F = \hat{f} \sim \text{Bernoulli}(\rho)$, $\ell | X = x \sim \text{Bernoulli}(\bar{\hat{f}}(x))$. As a result, the point-wise diversity can be written as

$$\text{Var}_{F \sim \hat{p}_F}(F(x)) = \mathbb{E}_{F \sim \hat{p}_F}[F(x)^2] - \mathbb{E}_{F \sim \hat{p}_F}[F(x)]^2 \tag{6}$$

$$= \mathbb{E}_{F \sim \hat{p}_F}[F(x)] - \mathbb{E}_{F \sim \hat{p}_F}[F(x)]^2 = \bar{\hat{f}}(x)(1 - \bar{\hat{f}}(x)) \tag{7}$$

This shows that for modern DNNs with neural collapse, the point-wise diversity is minimized either when most models make the correct prediction or when most models make the wrong prediction. Without considering the distributions $p_F \otimes p_X$, such strong correlation between point-wise diversity and point-wise prediction cannot be discovered and the analysis on point-wise diversity deviates from the practice. For example, Abe et al. (2022a) discovered that the point-wise diversity equals to the Jensen gap of the Brier score, which falls short due to the drawbacks discussed in section 3. It has been demonstrated in section 4.1 that $\ell | F = f$ are identical regardless of $f$. We thus assume the following condition throughout the derivations.

- **Distributional Equivalence**: $\forall f^{(i)}, f^{(j)} \in \mathcal{F}, P(\ell | F = f^{(i)}) = P(\ell | F = f^{(j)})$.

This leads to the distributional equivalence of $\hat{p}_F$: $\forall \hat{f}^{(i)}, \hat{f}^{(j)} \in \hat{\mathcal{F}}, P(\ell | F = \hat{f}^{(i)}) = P(\ell | F = \hat{f}^{(j)})$.

## 4.3 THE DISTRIBUTION OF DATA

According to the distributional equivalence across models, we note that the point-wise difference can be mitigated when we consider the entire data distribution $p_X$. All the models have the same expected predictions over the entire dataset. Based on this, we then prove the following theorem that explains the improvement of ensembles.

**Theorem 4.1.** *(Guaranteed Improvement) Given a convex metric $\phi$, we have:*

$$\mathbb{E}_{X \sim p_X}\left[\phi\left(\mathbb{E}_{F \sim \hat{p}_F}[F(X)]\right)\right] \leq \min_{\hat{f} \in \hat{\mathcal{F}}}\left\{\mathbb{E}_{X \sim p_X}\left[\phi(\hat{f}(X))\right]\right\} \tag{8}$$

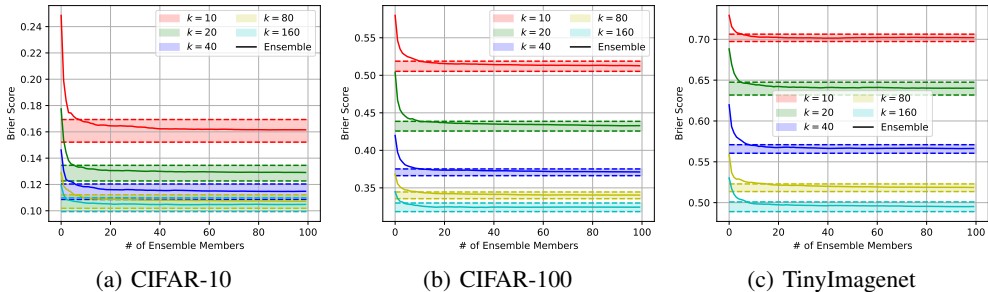

(a) CIFAR-10          (b) CIFAR-100          (c) TinyImagenet

Figure 4: The estimation (regions) and the empirical results (solid curves) of the Brier score of original models. The $x$-axis represents the number of ensemble members (a.k.a. ensemble capacity), and the $y$-axis represents the Brier score of the ensemble. Dashed lines represent the standard deviations. The tested model is CNN.

The proof is presented in appendix B.3. This shows the condition eq. (4) where deep ensembles are preferred under neural collapse. It demonstrates that deep ensembles always outperform **any** individual ensemble members (instead of the average loss).

## 4.4 THE FACTORS OF THE ENSEMBLE IMPROVEMENT

Although it has been proved above that deep ensembles are guaranteed to outperform *any single member*, the extent to which the improvement is and what factors contribute to the improvement remains unknown. Therefore, we delve deep into the quantification analysis of the improvement.

Instantiating $\phi$ to the Brier score $\phi_B$, we have the following theorem to bound the performance shift.

**Theorem 4.2.** *(The Brier Score Improvement) The ensemble performance improvement of the Brier score is determined by:*

$$\Delta_B = \min_{\hat{f}\in\hat{\mathcal{F}}}\mathbb{E}_{X\sim p_X}\left[\phi_B\left(\hat{f}(X)\right)\right] - \mathbb{E}_{X\sim p_X}\left[\phi_B\left(\mathbb{E}_{F\sim\hat{p}_F}[F(X)]\right)\right] = \rho - \rho^2 - \mathrm{Var}_{p_X}[\bar{\hat{f}}(X)] \quad (9)$$

*where $\rho = \mathbb{E}_{X\sim p_X}[\hat{f}(X)]$ for $\forall \hat{f}\in\hat{\mathcal{F}}$. $\Delta_B = 0$ holds if and only if point-wise equivalence holds. $\Delta_B = \rho - \rho^2$ holds if and only if the ensemble $\bar{\hat{f}}$ predicts constantly for all samples.*

The proof is deferred to appendix B.5. The result shows that when the expected prediction of $\hat{p}_F$ is determined (as $\rho$), then the **global diversity** $\mathrm{Var}_{X\sim p_X}[\bar{\hat{f}}(X)]$ is responsible for the improvement. To our best knowledge, previous work has been focusing on point-wise diversity (i.e. variance over $p_F$) while the importance of the global diversity (i.e. variance over $p_X$) has not been recognized before. Now it suffices to determine how the global diversity changes regarding the joint distribution $p_X \otimes p_F$. Recall from eq. (7) that if individual models tend to *agree* more on each sample, the point-wise predictions become closer to either 0 or 1. This will lead to a higher global diversity since the ensembled predictions are deviated from the center. Such agreement can be measured by *the probability measure of the shared correct prediction between two models.* In fact, We have the following theorem proved in appendix B.6:

**Theorem 4.3.** *Let $F^{-1}(1) = \{\boldsymbol{x} \in \mathcal{X}|F(\boldsymbol{x}) = 1\}$ denote the subset of $\mathcal{X}$ that $F$ can predict correctly, then $\mathrm{Var}_{X\sim p_X}[\bar{\hat{f}}(X)] = \mathbb{E}_{F_1,F_2\sim\hat{\mathcal{F}}}\left[p_X(F_1^{-1}(1)\cap F_2^{-1}(1))\right] - \rho^2$*

This quantifies that the global diversity of $\hat{p}_F$ decreases with $\rho$ and increases with the expected agreement across models. We emphasize that this is not counterintuitive since the globalness is regarding $p_X$ instead of $p_F$. One important significance of theorem 4.3 is that due to the distributional equivalence property, trained models $F \sim p_F$ demonstrate great symmetry, leading to very small variance in $p_X(F_1^{-1}(1)\cap F_2^{-1}(1))$. Thus it can be estimated by only two models with small errors. As a result, **we can foresee how the ensemble will perform with only two models**. Besides, due to the associative and commutative nature of multiplication, theorem 4.3 can be migrated to the original function space $\mathcal{F}$. We prove the following theorem in appendix B.7:

**Theorem 4.4.** *The Brier score of the ensemble $\bar{f}$ can be estimated by*

$$\mathbb{E}_{X\sim p_X}[\phi_B(\bar{f}(X))] = \mathbb{E}_{F_1,F_2\sim p_F}\left[\mathbb{E}_{X\sim p_X}[F_1(X)F_2(X)]\right] - 2\mathbb{E}_{F\sim p_F}[\mathbb{E}_{X\sim p_X}[F(X)]] + 1 \quad (10)$$

Due to the distributional equivalence across all models, we can estimate this by $f^{(1)}, f^{(2)} \in \mathcal{F}$:

$$\mathbb{E}_{X \sim p_X}[\phi_B(\bar{f}(X))] \approx \mathbb{E}_{X \sim p_X}[f^{(1)}(\boldsymbol{x})f^{(2)}(\boldsymbol{x})] - 2\mathbb{E}_{X \sim p_X}[f^{(1)}(\boldsymbol{x})] + 1 \quad (11)$$

This is verified empirically in fig. 4 by using all 9900 possible pairs to estimate the ensemble performance. The mean and standard deviation of the estimation are visualized as the regions, while the solid curves represent the loss of the ensembles. Therefore, we can use only **two** models to accurately estimate the asymptotic loss of deep ensembles. Using this scheme, practitioners can make judicious and efficient choices in the application of deep ensembles and balance the trade-off between training budget and performance requirements.

**Bias-Variance Trade-Off of Ensembles.** Another intriguing yet unsolved phenomenon of deep ensembles is its bias-variance trade-off as the single-model capacity of each ensemble member increases (Geiger et al., 2020; Kobayashi et al., 2021). Note that the ensemble performance under the Brier score can be written as a function of $\rho = \mathbb{E}_X[f(X)]$ for any $f \in \mathcal{F}$ as $S(\rho) = \mathbb{E}_X[\bar{f}(X)^2] - 2\mathbb{E}_X[\bar{f}(X)] + 1 = \rho^2 - 2\rho + 1 + \mathrm{Var}_X[\bar{f}(X)]$. From fig. 5, we know that $\mathrm{Var}_X[\bar{f}(X)]$ tend to increase with $\rho$. Thus the Brier score $S(\rho)$ starts increasing before $\rho$ reaches 1 (as the single-model capacity increases). This leads to the bias-variance trade-off observed for deep ensembles.

## 5 ENSEMBLES OF LIMITED BUDGETS.

In practice, an approximation with $M$ models is utilized. It has been observed in many existing works that the loss decreases as $M$ increases. This phenomenon has not been understood either. Previous work simply terms $M$ as the "ensemble capacity", which does not contribute to the understanding of the internal mechanism given the $M$ models are trained completely independently. It is also observed that increasing the ensemble capacity yields similar performance improvement compared with increasing the capacity of a single model, as shown in fig. 6 (More results are shown in appendix E).

Figure 5: An illustration of the relation between $\mathbb{E}_X[\bar{f}(X)]$ and $\mathrm{Var}_X[\bar{f}(X)]$. All six structure-dataset combinations are tested here. $\rho$ is controlled by the single-model capacity $k$ of ensemble members. For each curve, $\rho$ increases as the width parameter $k$.

To determine the efficiency of ensembles regarding $M$, and also to unveil the connection between single-model capacity and ensemble capacity, we prove the following theorem in appendix B.8:

**Theorem 5.1.** *The expected Brier score loss of $M$ ensemble members is determined by*

$$\mathbb{E}_{F_1, \cdots, F_M \sim \hat{p}_F}\left[\phi_B\left(\frac{1}{M}\sum_{i=1}^M F_i(X)\right)\right] = \mathbb{E}_{X \sim p_X}\left[\phi_B(\bar{\bar{f}}(X))\right] + \frac{\mathbb{E}_{X \sim p_X}\left[\mathrm{Var}_{F \sim \hat{p}_F}(F(\boldsymbol{x}))\right]}{M} \quad (12)$$

$$= \frac{M+1}{M}\mathbb{E}_{X \sim p_X}[\bar{\bar{f}}(X)^2] - \frac{\rho^2}{M} - 2\rho + 1 \quad (13)$$

*The expected NLL of $M$ ensemble members is estimated by*

$$\mathbb{E}_{F_1, \cdots, F_M \sim \hat{p}_F}\left[\phi_{nll}\left(\frac{1}{M}\sum_{i=1}^M F_i(X)\right)\right] \approx \mathbb{E}_{X \sim p_X}\left[\phi_{nll}(\bar{\bar{f}}(X))\right] + \mathbb{E}_{X \sim p_X}\left[\frac{1 - \bar{\bar{f}}(X)}{2\bar{\bar{f}}(X)}\right]\frac{1}{M} \quad (14)$$

This theorem explains the constantly observed phenomenon where increasing the number of ensembles results in improved performance. Demonstrated by eqs. (12) and (14), the loss can be decomposed to the loss of the convergence result of the population means and another positive term controlled by $M$. Thus the loss **decreases** in an inverse-proportional manner as we increase the number $M$ of ensemble members, i.e. the ensemble capacity. As $M \to \infty$, the loss converges to the global loss of the population ensemble. Note that all values are determined by $p_F$ except for the ensemble budget $M$. As a consequence, this theorem presents a closed-form expression for the performance of deep ensembles. On the other hand, as demonstrated in appendix D, scaling up a single model changes $\rho$ instead, i.e., the prediction of a single model. This theoretically verifies that both single-model capacity and ensemble capacity contribute to performance improvement but in

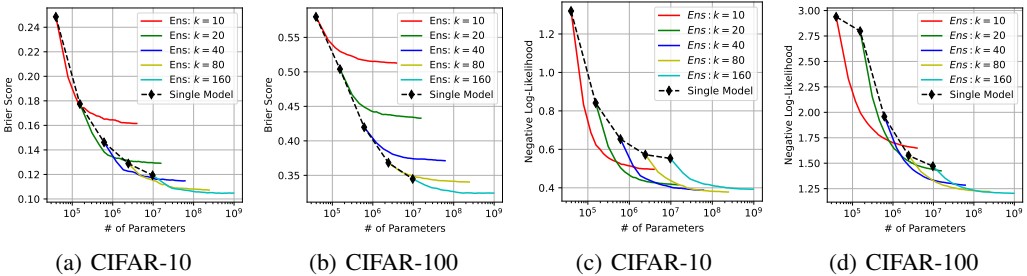

(a) CIFAR-10      (b) CIFAR-100      (c) CIFAR-10      (d) CIFAR-100

Figure 6: The performance comparison between the scaling of a single model (black dashed curves) and increasing the number of ensemble members (colorful solid curves). In each ensemble (i.e. each colorful curve), $M$ varies from 1 to 100. The results are generated using CNN models.

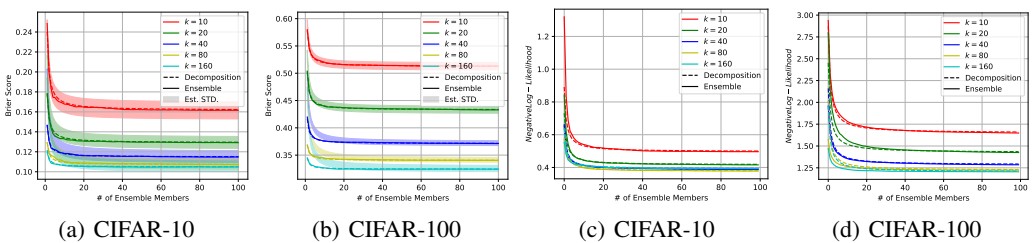

(a) CIFAR-10      (b) CIFAR-100      (c) CIFAR-10      (d) CIFAR-100

Figure 7: The verification of theorem 5.1. The theoretical results (dashed curves) are compared with the empirical results of ensembles using $M$ models for (a)(b) the Brier score and (c)(d) the negative log-likelihood. The ensemble curves are computed using $p_F$. Results are generated using CNNs.

completely distinct ways. We validate these theoretical results in fig. 7, where the empirical loss and the theoretical decomposition are compared. Note that although the theorems are derived for $\hat{\mathcal{F}}$ with complete neural collapse, it generalizes to $\mathcal{F}$ due to the already existing neural collapse nature in trained models. Therefore the experiments are also carried out for $\mathcal{F}$ to demonstrate the practical significance. It is observed that the curves almost completely overlap. The estimation of NLL deviates a little at the beginning due to Taylor's expansion approximation. Here we clip $(1 - \bar{\hat{f}}(X))/(2\bar{\hat{f}}(X))$ by an upper bound of 5 to avoid explosion caused by difficult samples to avoid overflow issues.

Besides, inspired by theorem 4.4, the Brier score can be rewritten as eq. (13), where the first term can be estimated by the *agreement* between two single models. We thus estimate the entire curve using only two models, and plot the standard deviation as the colorful regions in fig. 7(a)(b). This provides even further and deeper understanding compared with fig. 4.

## 6 CONCLUSIONS

In this work, we focus on the important problem of demystifying the "mysterious effectiveness" of deep ensembles. We diagnose and demonstrate the problem of existing ways of understanding deep ensembles and the inconsistency in practice. Then we reveal the distributional equivalence property of model distributions through comprehensive experimental results. Based on such a property, we provide theoretical analysis with rigorously proved theorems to (1) demonstrate the guarantee of deep ensembles' effectiveness; (2) propose a scheme to accurately estimate the asymptotic performance of infinitely many models using only two models; and (3) lucidly uncover the essence of increasing the ensemble capacity and scaling up a single model. In conclusion, our work discloses the true mechanism of mysterious deep ensembles for the first time. The derivations also provide valuable insights into the understanding of the behaviors of models through the perspective of the joint distribution between data and models. We admit that the distributional equivalence property is an observational conclusion. It remains an intriguing question why the probability measure of the testing data mastered by any single model retains an identical level across different models. We deduce it's due to certain properties of distribution shift and leave that for future explorations.

ACKNOWLEDGEMENTS

This work was partially supported by the EMBRIO Institute, contract #2120200, a National Science Foundation (NSF) Biology Integration Institute, and NSF IIS #1955890, IIS #2146091, IIS #2345235.

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

## A    EXPERIMENT SETUPS

To carry out empirical demonstration simultaneously with the theoretical analysis, we introduce the detailed experimental setup here. Experiments are carried out on Intel(R) Xeon(R) Gold 6226R CPU @ 2.90Hz with NVIDIA RTX A5000 GPUs.

The experiments are carried out on three datasets, CIFAR-10/100 (Krizhevsky et al., 2009) and TinyImagenet (Deng et al., 2009). For different model architectures, following a similar setup (Nakkiran et al., 2021), we consider both CNNs and ResNets(He et al., 2016). On the one hand, for each $p_F$, $M$ models are trained to obtain a comprehensive understanding of the influence of the ensemble size. The empirical studies of the properties of distributions of models require a large number of samples (i.i.d. trained models) due to the high nonlinearity and dimensions. However, in previous work, this has not been effectively addressed with a sufficiently large population (e.g. $M = 4$ in (Lee et al., 2015), $M = 5$ in (Abe et al., 2022b), $M = 14$ in (Fort et al., 2019), etc.). To study the distribution of a homogeneous ensemble of models, we train $M = 100$ models with distinct random seeds. On the other hand, note that increasing the ensemble size and increasing the capacity of single models both contribute to an improved performance. However, the relation between the mechanisms of these two distinct approaches remains unknown. Some empirical studies on the correlations even suggest that their mechanisms agree with each other. As a result, we set multiple different capacities for each architecture. Specifically, we use a factor $k$ to control the capacity through the width. CNNs contain a 4-layer convolutional layer with output channels $[k, 2k, 4k, 8k]$, each of which is followed by a batch normalization layer and a ReLU activation. We also add a max-pooling layer after the $2k, 4k, 8k$ convolutional layer, with strides 2, 2, 8. (For TinyImagenet, we use 2, 4, 8 instead.) Finally, a linear classification head takes the $8k$-dimensional embedding and maps it to the output space. Regarding ResNets, we scaled the width of ResNet-18 as suggested by (Nakkiran et al., 2021). The channel of each layer is linearly determined by $k$, where $k = 64$ results in the original ResNet-18. In order to explore the influence of model capacities, we train DNNs with $k \in K$, where $K = \{10, 20, 40, 80, 160\}$. Note that for single model capacities, the number of trainable parameters increases quadratically with respect to the factor $k$, while the trainable parameters of the entire ensemble increases only linearly with respect to the number of ensembles $M$.

As suggested by the essence of deep ensembles, SGD is used as the solver with no momentum or data augmentation. Let $t$ denote the number of epochs; we use the learning rate $\lambda(t) = \frac{\lambda_0}{\sqrt{t}}$, where $\lambda_0 = 0.1$ is the initial learning rate.

## B    PROOFS

### B.1    PROOF OF PROPOSITION 3.1

**Proposition 3.1.** $LG(\boldsymbol{x}) \leq 0$ always holds for any monotonic metric $\phi$. That is, given input sample $\boldsymbol{x}$, deep ensemble always underperforms the best individual member.

*Proof.* When $\phi$ is monotonically increasing. Then

$$\min_{f \in \mathcal{F}}\{\phi(f(\boldsymbol{x}))\} = \phi(\min_{f \in \mathcal{F}}\{f(\boldsymbol{x})\}) \tag{15}$$

Since $\min_{f \in \mathcal{F}}\{f(\boldsymbol{x})\} \leq \mathbb{E}_{F \sim p_F}[F(\boldsymbol{x})]$ and $\phi$ is monotonically increasing, we have

$$\phi(\min_{f \in \mathcal{F}}\{f(\boldsymbol{x})\}) \leq \phi(\mathbb{E}_{F \sim p_F}[F(\boldsymbol{x})]) \tag{16}$$

Combining them results in

$$\min_{f \in \mathcal{F}}\{\phi(f(\boldsymbol{x}))\} \leq \phi(\mathbb{E}_{F \sim p_F}[F(\boldsymbol{x})]) \tag{17}$$

Therefore

$$LG(\boldsymbol{x}) = \min_{f \in \mathcal{F}}\{\phi(f(\boldsymbol{x}))\} - \phi(\mathbb{E}_{F \sim p_F}[F(\boldsymbol{x})]) \leq 0 \tag{18}$$

In the opposite, when $\phi$ is monotonically decreasing. Then

$$\min_{f \in \mathcal{F}}\{\phi(f(\boldsymbol{x}))\} = \phi(\max_{f \in \mathcal{F}}\{f(\boldsymbol{x})\}) \tag{19}$$

Since $\max_{f \in \mathcal{F}} \{f(\boldsymbol{x})\} \geq \mathbb{E}_{F \sim p_F}[F(\boldsymbol{x})]$ and $\phi$ is monotonically decreasing, we have

$$\phi(\max_{f \in \mathcal{F}} \{f(\boldsymbol{x})\}) \leq \phi(\mathbb{E}_{F \sim p_F}[F(\boldsymbol{x})]) \tag{20}$$

Combining them results in

$$\min_{f \in \mathcal{F}} \{\phi(f(\boldsymbol{x}))\} \leq \phi(\mathbb{E}_{F \sim p_F}[F(\boldsymbol{x})]) \tag{21}$$

Therefore

$$LG(\boldsymbol{x}) = \min_{f \in \mathcal{F}} \{\phi(f(\boldsymbol{x}))\} - \phi(\mathbb{E}_{F \sim p_F}[F(\boldsymbol{x})]) \leq 0 \tag{22}$$

This proves the proposition.$\square$

### B.2 LEMMA B.1

For the model $\hat{f} \sim \hat{\mathcal{F}}$ with complete collapse, the prediction is either 0 or 1. Thus we have the following lemma:

**Lemma B.1.** *(Prediction Distributional Equivalence)* $\exists \rho \in [0, 1]$ s.t. $\forall \hat{f} \in \hat{\mathcal{F}}$, let $U = \{\boldsymbol{x} \in \mathcal{X} \subset \mathbb{R}^d | \hat{f}(\boldsymbol{x}) = 1\} = \hat{f}^{-1}(1)$, then $p_X(U) \equiv \rho$.

*Proof.* According to the distributional equivalence condition, for arbitrary $\hat{f}, \hat{g} \in \hat{\mathcal{F}}$, $P(\hat{f}(\boldsymbol{x})) = P(\hat{g}(\boldsymbol{x}))$ holds for $\forall \boldsymbol{x} \in \mathcal{X}$, and thus $\mathbb{E}_{X \sim p_X}[\hat{f}(X)] = \mathbb{E}_{X \sim p_X}[\hat{g}(X)]$. Therefore, $\exists \rho \in [0, 1]$ s.t. for an arbitrary $\hat{f} \in \hat{\mathcal{F}}$, $\mathbb{E}_{X \sim p_X}[\hat{f}(X)] = \rho$. Then

$$\int_{U_i} p_X(\boldsymbol{x}) \mathrm{d}\boldsymbol{x} = \int_{U_i} 1 \cdot p_X(\boldsymbol{x}) \mathrm{d}\boldsymbol{x} + \int_{\mathcal{X} \setminus U_i} 0 \cdot p_X(\boldsymbol{x}) \mathrm{d}\boldsymbol{x} \tag{23}$$

$$= \int_{\mathcal{X}} \hat{f}^{(i)}(\boldsymbol{x}) p_X(\boldsymbol{x}) \mathrm{d}\boldsymbol{x} = \mathbb{E}_{X \sim p_X}[\hat{f}^{(i)}(X)] = \rho \tag{24}$$

This proves the lemma.$\square$

### B.3 PROOF OF THEOREM 4.1

**Theorem 4.1** **(Guaranteed Improvement)** Given a convex metric $\phi$, we have:

$$\mathbb{E}_{X \sim p_X} \left[ \phi \left( \mathbb{E}_{F \sim \hat{p}_F}[F(X)] \right) \right] \leq \min_{\hat{f} \in \hat{\mathcal{F}}} \left\{ \mathbb{E}_{X \sim p_X} \left[ \phi(\hat{f}(X)) \right] \right\} \tag{25}$$

*Proof.* It suffices to show that for an arbitrary $\hat{g} \in \hat{\mathcal{F}}$,

$$\mathbb{E}_{X \sim p_X} \left[ \phi \left( \mathbb{E}_{F \sim \hat{p}_F}[F(X)] \right) \right] \leq \mathbb{E}_{X \sim p_X} \left[ \phi(\hat{g}(X)) \right] \tag{26}$$

Note that according to theorem B.1

$$RHS = \int_{\mathcal{X}} \phi(\hat{g}(\boldsymbol{x})) p_X(\boldsymbol{x}) \mathrm{d}\boldsymbol{x} \tag{27}$$

$$= \int_{U} p_X(\boldsymbol{x}) \phi(1) \mathrm{d}\boldsymbol{x} + \int_{V} p_X(\boldsymbol{x}) \phi(0) \mathrm{d}\boldsymbol{x} \tag{28}$$

$$= \phi(1)\rho + \phi(0)(1 - \rho) \tag{29}$$

$$\geq \phi(1 \cdot \rho + 0 \cdot (1 - \rho)) = \phi(\rho) \tag{30}$$

On the other hand, the LHS can be written as

$$LHS = \mathbb{E}_{X \sim p_X} \left[ \phi \left( \mathbb{E}_{F \sim \hat{p}_F}[F(X)] \right) \right] \tag{31}$$

$$\leq \phi \left( \mathbb{E}_{X \sim p_X} \left[ \mathbb{E}_{F \sim \hat{p}_F}[F(X)] \right] \right) \tag{32}$$

$$= \phi \left( \mathbb{E}_{F \sim \hat{p}_F} \left[ \mathbb{E}_{X \sim p_X}[\hat{f}(X)] \right] \right) = \phi(\rho) = RHS \tag{33}$$

Therefore, the statement is proven.$\square$

This result shows that for any individual model $f$ drawn from $p_F$, subspace $U$ of correctly predicted samples have the probability measure $\rho$ under $p_X$. And such a measure is irrelevant to $f$ or $U$. Instead, it is determined by the global view of $p_X \otimes p_F$.

### B.4 LEMMA B.2

Distributional equivalence ensures not only the identical mean prediction but also identical average loss under any metric $\phi$. Formally, we have:

**Lemma B.2.** *(Performance Equivalence under Metrics)* $\forall \phi \in \mathcal{C}([0,1])$, *then (i)* $\forall \hat{f}, \hat{g} \in \hat{\mathcal{F}}$, $\mathbb{E}_{X \sim p_X}[\phi(\hat{f}(X))] = \mathbb{E}_{X \sim p_X}[\phi(\hat{g}(X))]$. *(ii)* $\forall f, g \in \mathcal{F}$, $\mathbb{E}_{X \sim p_X}[\phi(f(X))] = \mathbb{E}_{X \sim p_X}[\phi(g(X))]$.

*Proof.* (i) The equivalence of performance for two random models $\hat{f}, \hat{g}$ with complete collapse can be proved as:

$$\mathbb{E}_{X \sim p_X}[\phi(\hat{f}(X))] = \int_{\mathcal{X}} \phi(\hat{f}(\boldsymbol{x}))p_X(\boldsymbol{x})\mathrm{d}\boldsymbol{x} \tag{34}$$

$$= \int_{U_f} p_X(\boldsymbol{x})\phi(1)\mathrm{d}\boldsymbol{x} + \int_{\mathcal{X} \setminus U_f} p_X(\boldsymbol{x})\phi(0)\mathrm{d}\boldsymbol{x} \tag{35}$$

$$= \rho\phi(1) + (1-\rho)\phi(0) \tag{36}$$

$$= \int_{U_g} p_X(\boldsymbol{x})\phi(1)\mathrm{d}\boldsymbol{x} + \int_{\mathcal{X} \setminus U_g} p_X(\boldsymbol{x})\phi(0)\mathrm{d}\boldsymbol{x} \tag{37}$$

$$= \int_{\mathcal{X}} \phi(\hat{g}(\boldsymbol{x}))p_X(\boldsymbol{x})\mathrm{d}\boldsymbol{x} = \mathbb{E}_{X \sim p_X}[\phi(\hat{g}(X))] \tag{38}$$

(ii) Let $f, g \in \mathcal{F}$ be two arbitrary model. According to the distributional equivalence property, we have $P(\ell|F = f) = P(\ell|F = g)$. Let $\mu_f$ be a pushforward measure such that $\forall I \subseteq [0,1]$, $\mu_f(I) = P(f(V) \in I)$, $\mu_g$ is defined similarly. According to the distributional equivalence condition, $\forall I \in [0,1], \mu_f(I) = \mu_g(I)$. Therefore:

$$\mathbb{E}_{X \sim p_X}[\phi(f(X))] = \int_{\mathcal{X}} \phi(f(\boldsymbol{x}))p_X(\boldsymbol{x})\mathrm{d}\boldsymbol{x} \tag{39}$$

$$= \int_0^1 \phi(t)\mu_f(\mathrm{d}t) \tag{40}$$

$$= \int_0^1 \phi(t)\mu_g(\mathrm{d}t) = \mathbb{E}_{X \sim p_X}[\phi(g(X))] \tag{41}$$

$$\square$$

As a result, all individual models have identical performance under arbitrary metric $\phi$.

### B.5 PROOF OF THEOREM 4.2

**Theorem 4.2** **(The Brier Score Improvement)** The ensemble performance improvement of the Brier score is tightly bounded by:

$$0 \le \min_{\hat{f} \in \hat{\mathcal{F}}} \mathbb{E}_{X \sim p_X}\left[\phi_B\big(\hat{f}(X)\big)\right] - \mathbb{E}_{X \sim p_X}\left[\phi_B\big(\mathbb{E}_{F \sim \hat{p}_F}[F(X)]\big)\right] \le \rho - \rho^2 \tag{42}$$

where $\rho = \mathbb{E}_{X \sim p_X}[\hat{f}(X)]$ for $\forall \hat{f} \in \hat{\mathcal{F}}$. $\Delta_B = 0$ holds if and only if point-wise equivalence holds. $\Delta_B = \rho - \rho^2$ holds if and only if the ensemble $\bar{\hat{f}}$ predicts constantly for all samples. Besides, $\mathrm{Var}_{X \sim p_X}[\bar{\hat{f}}(X)]$ is responsible for the performance improvement.

*Proof.* From theorem B.2, all individual models have identical performance. Thus $\forall \hat{f} \in \hat{\mathcal{F}}$,

$$\min_{\hat{g} \in \hat{\mathcal{F}}} \mathbb{E}_{X \sim p_X}[\phi(\hat{g}(X))] = \mathbb{E}_{X \sim p_X}[\phi(\hat{f}(X))] \tag{43}$$

Therefore, the ensemble performance improvement in Brier score is

$$\Delta_B = \mathbb{E}_{X \sim p_X}[\phi(\hat{f}(X))] - \mathbb{E}_{X \sim p_X}[\phi(\mathbb{E}_{F \sim \hat{p}_F}[F(X)])] \tag{44}$$

$$= (1 - \rho)\phi(0) + \rho\phi(1) - \mathbb{E}_{X \sim p_X}[(1 - \bar{\hat{f}}(X))^2] \tag{45}$$

$$= 1 - \rho - \mathbb{E}_{X \sim p_X}[\bar{\hat{f}}(\boldsymbol{x})^2 - 2\bar{\hat{f}}(\boldsymbol{x}) + 1] \tag{46}$$

$$= 2\mathbb{E}_{X \sim p_X}[\bar{\hat{f}}(\boldsymbol{x})] - \mathbb{E}_{X \sim p_X}[\bar{\hat{f}}(\boldsymbol{x})^2] - \rho \tag{47}$$

$$= 2\mathbb{E}_{X \sim p_X}[\bar{\hat{f}}(\boldsymbol{x})] - \mathbb{E}_{X \sim p_X}[\bar{\hat{f}}(\boldsymbol{x})^2] - \mathbb{E}_{X \sim p_X}[\bar{\hat{f}}(\boldsymbol{x})] \tag{48}$$

$$= \mathbb{E}_{X \sim p_X}[\bar{\hat{f}}(\boldsymbol{x})] - \mathbb{E}_{X \sim p_X}[\bar{\hat{f}}(\boldsymbol{x})^2] \tag{49}$$

$$= \mathbb{E}_{X \sim p_X}[\bar{\hat{f}}(\boldsymbol{x})(1 - \bar{\hat{f}}(\boldsymbol{x}))] \geq 0 \tag{50}$$

This proves the lower bound. And the lower bound holds if and only if $\forall \boldsymbol{x} \in \mathcal{X}, \bar{\hat{f}}(\boldsymbol{x})(1 - \bar{\hat{f}}(\boldsymbol{x})) \equiv 0$. This means that for any $\boldsymbol{x} \in \mathcal{X}, \bar{\hat{f}}(\boldsymbol{x})$ is either 0 or 1. As a result, $\forall \boldsymbol{x} \in \mathcal{X}$, either $\forall \hat{f} \in \hat{\mathcal{F}}, \hat{f}(\boldsymbol{x}) = 0$ or $\forall \hat{f} \in \hat{\mathcal{F}}, \hat{f}(\boldsymbol{x}) = 1$. Therefore, the point-wise equivalence hold.

As for the upper bound, note that $\mathbb{E}_{X \sim p_X}[\bar{\hat{f}}(\boldsymbol{x})] = \rho$, then

$$\Delta_B = \mathbb{E}_{X \sim p_X}[\bar{\hat{f}}(\boldsymbol{x})] - \mathbb{E}_{X \sim p_X}[\bar{\hat{f}}(\boldsymbol{x})]^2 + \mathbb{E}_{X \sim p_X}[\bar{\hat{f}}(\boldsymbol{x})]^2 - \mathbb{E}_{X \sim p_X}[\bar{\hat{f}}(\boldsymbol{x})^2] \tag{51}$$

$$= \mathbb{E}_{X \sim p_X}[\bar{\hat{f}}(\boldsymbol{x})] - \mathbb{E}_{X \sim p_X}[\bar{\hat{f}}(\boldsymbol{x})]^2 - \mathrm{Var}_{X \sim p_X}[\bar{\hat{f}}(\boldsymbol{x})] \tag{52}$$

$$= \rho - \rho^2 - \mathrm{Var}_{X \sim p_X}[\bar{\hat{f}}(\boldsymbol{x})] \tag{53}$$

Since $\mathrm{Var}_{X \sim p_X}[\bar{\hat{f}}(\boldsymbol{x})] \geq 0$, we have $\Delta_B \leq \rho - \rho^2$, and the equality holds if and only if $\mathrm{Var}_{X \sim p_X}[\bar{\hat{f}}(\boldsymbol{x})] \equiv 0$. That is, the ensemble predicts constantly for all input samples. Thus the theorem is proved. $\square$

### B.6 PROOF OF THEOREM 4.3

**Theorem 4.3** Let $F^{-1}(1) = \{\boldsymbol{x} \in \mathcal{X} | F(\boldsymbol{x}) = 1\}$ denote the subset of $\mathcal{X}$ that $F$ can predict correctly, then $\mathrm{Var}_{X \sim p_X}[\bar{\hat{f}}(X)] = \mathbb{E}_{F_1, F_2 \sim \hat{\mathcal{F}}}[p_X(F_1^{-1}(1) \cap F_2^{-1}(1))] - \rho^2$

*Proof.* By definition, the global diversity (LHS) can be written as follows:

$$\mathrm{Var}_{X \sim p_X}[\bar{\hat{f}}(X)] = \mathbb{E}_{X \sim p_X}[\bar{\hat{f}}(X)^2] - \mathbb{E}_{X \sim p_X}[\bar{\hat{f}}(X)]^2 \tag{54}$$

$$= \mathbb{E}_{X \sim p_X}[\bar{\hat{f}}(X)^2] - \rho^2 \tag{55}$$

Thus it suffices to show that $\mathbb{E}_{X \sim p_X}[\bar{\hat{f}}(X)^2] = \mathbb{E}_{F_1, F_2 \sim \hat{p}_F}[p_X(F_1^{-1}(1) \cap F_2^{-1}(1))]$. Starting from the RHS, the expectation of the probability measure of the intersection can be re-written as:

$$\mathbb{E}_{F_1, F_2 \sim \hat{p}_F}[p_X(F_1^{-1}(1) \cap F_2^{-1}(1))] = \mathbb{E}_{F_1, F_2 \sim \hat{p}_F}\left[\int_{F_1^{-1}(1) \cap F_2^{-1}(1)} p_X(\boldsymbol{x})\mathrm{d}\boldsymbol{x}\right] \tag{56}$$

Note that $F_1^{-1}(1) = \{\boldsymbol{x} \in \mathcal{X} | F_1(\boldsymbol{x}) = 1\}, F_2^{-1}(1) = \{\boldsymbol{x} \in \mathcal{X} | F_2(\boldsymbol{x}) = 1\}$, we have

$$F_1^{-1}(1) \cap F_2^{-1}(1) = \{\boldsymbol{x} \in \mathcal{X} | F_1 = 1, F_2 = 1\} \tag{57}$$

$$= \{\boldsymbol{x} \in \mathcal{X} | F_1(\boldsymbol{x})F_2(\boldsymbol{x}) = 1\} \tag{58}$$

Thus according to Fubini's Theorem:

$$\mathbb{E}_{F_1,F_2\sim\hat{p}_F}\Big[p_X(F_1^{-1}(1)\cap F_2^{-1}(1))\Big] = \mathbb{E}_{F_1,F_2\sim\hat{p}_F}\Big[\int_{\mathcal{X}}F_1(\boldsymbol{x})F_2(\boldsymbol{x})p_X(\boldsymbol{x})\mathrm{d}\boldsymbol{x}\Big] \quad (59)$$

$$= \int_{\mathcal{X}}\Big(\mathbb{E}_{F_1,F_2\sim\hat{p}_F}\big[F_1(\boldsymbol{x})F_2(\boldsymbol{x})\big]\Big)p_X(\boldsymbol{x})\mathrm{d}\boldsymbol{x} \quad (60)$$

$$= \int_{\mathcal{X}}\Big(\mathbb{E}_{F\sim\hat{p}_F}[F(\boldsymbol{x})]^2\Big)p_X(\boldsymbol{x})\mathrm{d}\boldsymbol{x} \quad (61)$$

$$= \int_{\mathcal{X}}\bar{\hat{f}}(\boldsymbol{x})^2 p_X(\boldsymbol{x})\mathrm{d}\boldsymbol{x} \quad (62)$$

$$= \mathbb{E}_{X\sim p_X}[\bar{\hat{f}}(\boldsymbol{x})^2] \quad (63)$$

Thus the theorem is proved. $\qquad\square$

### B.7 PROOF OF THEOREM 4.4

**Theorem 4.4** The Brier score of the ensemble $\bar{f}$ can be estimated by

$$\mathbb{E}_{X\sim p_X}[\phi_B(\bar{f}(X))] = \mathbb{E}_{F_1,F_2\sim p_F}\big[\mathbb{E}_{X\sim p_X}[F_1(X)F_2(X)]\big] - 2\mathbb{E}_{F\sim p_F}\big[\mathbb{E}_{X\sim p_X}[F(X)]\big] + 1 \quad (64)$$

*Proof.* The ensemble performance can be written as

$$\mathbb{E}_{X\sim p_X}[\phi_B(\bar{f}(X)] = \mathbb{E}_{X\sim p_X}\Big[\big(1-\bar{f}(X)\big)^2\Big] \quad (65)$$

$$= \mathbb{E}_{X\sim p_X}\big[\mathbb{E}_{F\sim p_F}[F(X)]^2\big] - 2\mathbb{E}_{X\sim p_X}\big[\mathbb{E}_{F\sim p_F}[F(X)]\big] + 1 \quad (66)$$

$$= \mathbb{E}_{X\sim p_X}\big[\mathbb{E}_{F_1,F_2\sim p_F}[F_1(X)F_2(X)]\big] - 2\mathbb{E}_{F\sim p_F}\big[\mathbb{E}_{X\sim p_X}[F(X)]\big] + 1 \quad (67)$$

$$= \mathbb{E}_{F_1,F_2\sim p_F}\big[\mathbb{E}_{X\sim p_X}[F_1(X)F_2(X)]\big] - 2\mathbb{E}_{F\sim p_F}\big[\mathbb{E}_{X\sim p_X}[F(X)]\big] + 1 \quad (68)$$

Thus the theorem is proved. $\qquad\square$

### B.8 PROOF OF THEOREM 5.1

**Theorem 5.1** The expected Brier score loss of $M$ ensemble members is determined by

$$\mathbb{E}_{F_1,\cdots,F_M\sim\hat{p}_F}\Big[\phi_B\big(\frac{1}{M}\sum_{i=1}^{M}F_i(X)\big)\Big] = \frac{M+1}{M}\mathbb{E}_{X\sim p_X}[\bar{\hat{f}}(X)^2] - \frac{\rho^2}{M} - 2\rho + 1 \quad (69)$$

The expected NLL of $M$ ensemble members is estimated by

$$\mathbb{E}_{F_1,\cdots,F_M\sim\hat{p}_F}\Big[\phi_B\big(\frac{1}{M}\sum_{i=1}^{M}F_i(X)\big)\Big] \approx \mathbb{E}_{X\sim p_X}\big[\phi_{nll}(\bar{\hat{f}}(X))\big] + \mathbb{E}_{X\sim p_X}\Big[\frac{1-\bar{\hat{f}}(X)}{2\hat{f}(X)}\Big]\frac{1}{M} \quad (70)$$

*Proof.* Let $U_i = \{\boldsymbol{x}\in\mathcal{X}|\hat{f}^{(i)}(\boldsymbol{x})=1\} = p_X\big((\hat{f}^{(i)})^{-1}(1)\big)$. And define an indicate $\mathcal{I}_i(\boldsymbol{x})$ as

$$\mathcal{I}_i(\boldsymbol{x}) = \begin{cases} 1 & \text{if } \boldsymbol{x}\in U_i \\ 0 & \text{if } \boldsymbol{x}\notin U_i \end{cases} \quad (71)$$

Then the ensemble of $M$ members can be written as

$$\mathbb{E}_{X\sim p_X}\Big[\phi\big(\frac{1}{M}\sum_{i=1}^{M}\hat{f}^{(i)}(X)\big)\Big] = \sum_{m=0}^{M}\int_{\Omega_m}\phi\big(\frac{m}{M}\big)p_X(\boldsymbol{x})\mathrm{d}\boldsymbol{x} = \sum_{m=0}^{M}\phi\big(\frac{m}{M}\big)p_X(\Omega_m) \quad (72)$$

where $\Omega_i \subset \mathcal{X}$ such that

$$\Omega_m = \Big\{\boldsymbol{x}\in\mathcal{X}\big|\sum_{i=1}^{M}\mathcal{I}_i(\boldsymbol{x})=m\Big\} \quad (73)$$

That is, the set of samples that exactly $m$ members predict correctly.

Consider $M$ i.i.d. models $F_1, \cdots, F_M \sim \hat{p}_F$, we notice that $\sum_{i=1}^{M} \mathcal{I}_i(\boldsymbol{x}) \sim \text{Binomial}(M, \bar{\hat{f}}(\boldsymbol{x}))$. Thus $P(\sum_{i=1}^{M} \mathcal{I}_i(\boldsymbol{x}) = m) = \binom{M}{m} \bar{\hat{f}}(\boldsymbol{x})^m (1 - \bar{\hat{f}}(\boldsymbol{x}))^{M-m}$. As a result, the expected performance of an ensemble of $M$ models can be written as

$$\mathbb{E}_{F_1, \cdots, F_M \sim \hat{p}_F} \left[ \mathbb{E}_{X \sim p_X} \left[ \phi\left( \frac{1}{M} \sum_{i=1}^{M} \hat{f}^{(i)}(X) \right) \right] \right] \tag{74}$$

$$= \mathbb{E}_{F_1, \cdots, F_M \sim \hat{p}_F} \left[ \sum_{m=0}^{M} \phi\left( \frac{m}{M} \right) p_X(\Omega_i) \right] \tag{75}$$

$$= \sum_{m=0}^{M} \phi\left( \frac{m}{M} \right) \mathbb{E}_{F_1, \cdots, F_M \sim \hat{p}_F} \left[ p_X(\Omega_m) \right] \tag{76}$$

$$= \sum_{m=0}^{M} \phi\left( \frac{m}{M} \right) \mathbb{E}_{X \sim p_X} \left[ \binom{M}{m} \bar{\hat{f}}(X)^m (1 - \bar{\hat{f}}(X))^{M-m} \right] \tag{77}$$

$$= \mathbb{E}_{X \sim p_X} \left[ \sum_{m=0}^{M} \phi\left( \frac{m}{M} \right) \binom{M}{m} \bar{\hat{f}}(X)^m (1 - \bar{\hat{f}}(X))^{M-m} \right] \tag{78}$$

$$= \mathbb{E}_{X \sim p_X} \left[ E_{Z \sim \text{Binomial}(M, \bar{\hat{f}}(X))} \left[ \phi\left( \frac{Z}{M} \right) \right] \right] \tag{79}$$

For Brier score, this can be written as

$$\mathbb{E}_{X \sim p_X} \left[ \mathbb{E}_Z \left[ \left( 1 - \frac{Z}{M} \right)^2 \right] \right] = \mathbb{E}_Z \left[ 1 - \frac{2Z}{M} + \left( \frac{Z}{M} \right)^2 \right] \tag{80}$$

$$= \mathbb{E}_{X \sim p_X} \left[ 1 - 2\bar{\hat{f}}(X) + \frac{M\bar{\hat{f}}(X)(1 - \bar{\hat{f}}(X)) + M^2 \bar{\hat{f}}(X)^2}{M^2} \right] \tag{81}$$

$$= \mathbb{E}_{X \sim p_X} \left[ (1 - \bar{\hat{f}}(X))^2 + \frac{\bar{\hat{f}}(X)(1 - \bar{\hat{f}}(X))}{M} \right] \tag{82}$$

$$= \mathbb{E}_{X \sim p_X} \left[ (1 - \bar{\hat{f}}(X))^2 \right] + \frac{\mathbb{E}_{X \sim p_X} \left[ \bar{\hat{f}}(X)(1 - \bar{\hat{f}}(X)) \right]}{M} \tag{83}$$

$$= \mathbb{E}_{X \sim p_X} \left[ \phi_B(\bar{\hat{f}}(X)) \right] + \frac{\mathbb{E}_{X \sim p_X} \left[ \text{Var}_{F \sim \hat{p}_F}(F(\boldsymbol{x})) \right]}{M} \tag{84}$$

Note that

$$\mathbb{E}_{X \sim p_X}[\phi_B(\bar{\hat{f}}(X))] = \mathbb{E}_{X \sim p_X}[\bar{\hat{f}}(X)^2] - 2\rho + 1 \tag{85}$$

and from theorem 4.3, we also have

$$\text{Var}_{X \sim p_X}[\bar{\hat{f}}(X)] = \mathbb{E}_{X \sim p_X}[\bar{\hat{f}}(X)^2] - \rho^2 \tag{86}$$

Thus the expected performance can be written as

$$\mathbb{E}_{X \sim p_X} \left[ \mathbb{E}_Z \left[ \left( 1 - \frac{Z}{M} \right)^2 \right] \right] = \mathbb{E}_{X \sim p_X} \left[ \phi_B(\bar{\hat{f}}(X)) \right] + \frac{\mathbb{E}_{X \sim p_X} \left[ \text{Var}_{F \sim \hat{p}_F}(F(\boldsymbol{x})) \right]}{M} \tag{87}$$

$$= \mathbb{E}_{X \sim p_X}[\bar{\hat{f}}(X)^2] - 2\rho + 1 + \frac{\mathbb{E}_{X \sim p_X}[\bar{\hat{f}}(X)^2] - \rho^2}{M} \tag{88}$$

$$= \frac{M+1}{M} \mathbb{E}_{X \sim p_X}[\bar{\hat{f}}(X)^2] - \frac{\rho^2}{M} - 2\rho + 1 \tag{89}$$

For negative log-likelihood, we have

$$E_{X \sim p_X} \left[ E_{Z \sim \text{Binomial}(M, \bar{\hat{f}}(X))} \left[ \phi\left( \frac{Z}{M} \right) \right] \right] = \mathbb{E}_{X \sim p_X} \left[ \mathbb{E}_Z \left[ -\log\left( \frac{Z}{M} \right) \right] \right] \tag{90}$$

Using Taylor's expansion around $Z \approx M\bar{\hat{f}}(X)$, we have

$$\log\left(\frac{Z}{M}\right) \approx \log(\bar{\hat{f}}(X)) + \frac{Z - M\bar{\hat{f}}(X)}{M\bar{\hat{f}}(X)} - \frac{(Z - M\bar{\hat{f}}(X))^2}{(M\bar{\hat{f}}(X))^2} \tag{91}$$

Taking the expecation over $Z$, we have

$$\mathbb{E}_{X\sim p_X}\left[\mathbb{E}_Z\left[-\log\left(\frac{Z}{M}\right)\right]\right] = \mathbb{E}_{X\sim p_X}\left[\phi_{nll}(\bar{\hat{f}}(X)) - \frac{0}{M\bar{\hat{f}}(X)} + \frac{M\bar{\hat{f}}(X)(1 - \bar{\hat{f}}(X))}{(M\bar{\hat{f}}(X))^2}\right] \tag{92}$$

$$= \mathbb{E}_{X\sim p_X}\left[\phi_{nll}(\bar{\hat{f}}(X))\right] + \mathbb{E}_{X\sim p_X}\left[\frac{1 - \bar{\hat{f}}(X)}{2\bar{\hat{f}}(X)}\right]\frac{1}{M} \tag{93}$$

Thus the theorem is proved. $\qquad\square$

## B.9 PROOF OF THEOREM 5.1 WITHOUT SURROGATE MODELS $\hat{\mathcal{F}}$

Here we present a brief analysis of theorem 5.1 as complementary to appendix B.8. We show that the original distribution $p_F$ defined over the function space $\mathcal{F}$ also satisfies a more generalized version of the theorem compared with $\hat{p}_F$. We also demonstrate why the surrogate $\hat{p}_F$ is important in the proof. For any $X \sim p_X$, we have a distribution of the prediction of the ensemble: $\bar{f}(X) \sim p_\mu$ where $\bar{f}(\boldsymbol{x}) = \mathbb{E}_{X\sim p_X}[\mathbb{E}_{F\sim p_F}[F(X)]]$. Then instead of the discrete $\Omega_m$, we have infinitely many $\Omega_r = \{\boldsymbol{x} \in \mathcal{X} | \frac{1}{M}\sum_{i=1}^M f^{(i)}(\boldsymbol{x}) = r\}$ for $r \in [0, 1]$. And thus $p_X(\Omega_r) = p_\mu(r)$.

As a result, the expected loss can be written as

$$\mathbb{E}_{X\sim p_X}\left[\phi\left(\frac{1}{M}\sum_{i=1}^M F_i(X)\right)\right] = \int_0^1 \int_{\Omega_r} \phi(r)p_X(\boldsymbol{x})\mathrm{d}\boldsymbol{x}\mathrm{d}r \tag{94}$$

$$= \int_0^1 \phi(r)p_X(\Omega_r)\mathrm{d}r \tag{95}$$

Taking the expectation over i.i.d. $F_1, \cdots, F_M \sim p_F$, we have

$$\mathbb{E}_{F_1,\cdots,F_M\sim p_F}\mathbb{E}_{X\sim p_X}\left[\phi\left(\frac{1}{M}\sum_{i=1}^M F_i(X)\right)\right] = \mathbb{E}_{F_1,\cdots,F_M\sim p_F}\left[\int_0^1 \phi(r)p_X(\Omega_r)\mathrm{d}r\right] \tag{96}$$

$$= \int_0^1 \phi(r)\mathbb{E}_{F_1,\cdots,F_M\sim p_F}[p_X(\Omega_r)]\mathrm{d}r \tag{97}$$

It suffices to determine $\mathbb{E}_{F_1,\cdots,F_M\sim p_F}[p_X(\Omega_r)]$. Note that for any single model $f \in \mathcal{F}$ and $X \sim p_X$, we denote by $p_{\ell,f} = p_\ell$ the density function of $f(X)$. According to the distribution equivalence property, such a PDF is shared across all functions in $\mathcal{F}$. Therefore, we have

$$\mathbb{E}_{F_1,\cdots,F_M\sim p_F}[p_X(\Omega_r)] = \mathbb{E}_{F_1,\cdots,F_M\sim p_F}\left[\int_{\Omega_r} p_X(\boldsymbol{x})\mathrm{d}\boldsymbol{x}\right] \tag{98}$$

$$= \mathbb{E}_{F_1,\cdots,F_M\sim p_F}\left[\int_{\mathcal{X}} \mathbb{1}_{\frac{1}{M}\{\sum_{m=1}^M F_i(\boldsymbol{x})=r\}}p_X(\boldsymbol{x})\mathrm{d}\boldsymbol{x}\right] \tag{99}$$

$$= \int_{\mathcal{X}} \mathbb{E}_{F_1,\cdots,F_M\sim p_F}\left[\mathbb{1}_{\frac{1}{M}\{\sum_{m=1}^M F_i(\boldsymbol{x})=r\}}\right]p_X(\boldsymbol{x})\mathrm{d}\boldsymbol{x} \tag{100}$$

Consider that for a given $\boldsymbol{x}$ and $F \sim p_F$, the random variable $F(\boldsymbol{x})$ follows a distribution $p_{\ell,\boldsymbol{x}}$. And let $p_{M,\boldsymbol{x}}$ denote the distribution of the average of the distributions of $\frac{1}{M}\sum_{i=1}^M F_i(\boldsymbol{x})$. Then

$$\mathbb{E}_{F_1,\cdots,F_M\sim p_F}\left[\mathbb{1}_{\{\frac{1}{M}\sum_{m=1}^M F_i(\boldsymbol{x})=r\}}\right] = p_{M,\boldsymbol{x}}(r) \tag{101}$$

And then

$$\mathbb{E}_{F_1,\cdots,F_M\sim p_F}[p_X(\Omega_r)] = \int_{\mathcal{X}} p_{M,\boldsymbol{x}}(r)p_X(\boldsymbol{x})\mathrm{d}\boldsymbol{x} \tag{102}$$

$$= \mathbb{E}_{X\sim p_X}[p_{M,X}(r)] \tag{103}$$

Substitute this back to eq. (97), we have

$$\mathbb{E}_{X \sim p_X}\Big[\phi\big(\frac{1}{M}\sum_{i=1}^{M} F_i(X)\big)\Big] = \int_0^1 \phi(r)\mathbb{E}_{X \sim p_X}\big[p_{M,X}(r)\big]\mathrm{d}r \tag{104}$$

$$= \mathbb{E}_{X \sim p_X}\Big[\int_0^1 \phi(r) M(p_{\ell,X} * \cdots * p_{\ell,X})(Mr)\mathrm{d}r\Big] \tag{105}$$

where $(p_{\ell,X} * \cdots * p_{\ell,X})$ is the $M$-fold convolution of $p_{\ell,X}$. Now we abuse the notation and denote $m = Mr \in [0, M]$, then

$$\mathbb{E}_{X \sim p_X}\Big[\phi\big(\frac{1}{M}\sum_{i=1}^{M} F_i(X)\big)\Big] = \mathbb{E}_{X \sim p_X}\Big[\int_0^M \phi\big(\frac{m}{M}\big)(p_{\ell,X} * \cdots * p_{\ell,X})(m)\mathrm{d}m\Big] \tag{106}$$

$$= \mathbb{E}_{X \sim p_X}\Big[\mathbb{E}_{m \sim (p_{\ell,X} * \cdots * p_{\ell,X})}\big[\phi\big(\frac{m}{M}\big)\big]\Big] \tag{107}$$

For models with neural collapse, we have $p_{\ell,X} \approx \text{Bernoulli}$. This naturally leads to the previous results with $\hat{p}_F$ in eq. (79). And since $p_{\ell,X}$ is already very close to Bernoulli distribution, theorem 5.1 the conclusions of theorem 5.1 transfer to $p_F$ without requiring any additional modifications. This also shows the use of $\hat{p}_F$ is purely to avoid the retractable continuous distributions — If $p_{\ell,X}$ is modeled as a continuous distribution whose mass is mostly located at $\ell = 0$ and $\ell = 1$, the $M$-fold convolution becomes infeasible to track for closed-form analysis.

## B.10 TARGET-CLASS BRIER SCORES

The use of the output of the target class is based on the following reasons: (1) Because of the neural collapse phenomenon, the predicted probability of the target class is either very close to 1 or very close to 0. When it's close to 1, the prediction of all other classes is naturally zeros. And when it's close to 0, there is usually one other dominant class. Therefore, taking the target class only is already sufficient for the analysis. (2) Negative log-likelihood only considers the target class. Therefore, it creates unnecessary inconsistency across the analysis. (3) Due to the softmax layer, the output of the target class already contains information from the prediction of **all** classes (Wang & Wang, 2022).

Specifically, discriminative models do not directly predict the likelihood vector $\boldsymbol{p} \in [0, 1]^c$. Instead, they predict the logits $\boldsymbol{y} \in \mathbb{R}^c$, and then a softmax activation is applied to $\boldsymbol{y}$ to obtain the likelihood. Here $c$ is the number of classes. As a result, let $t$ denote the ground truth class, then the target-class Brier score is

$$(1 - p_t)^2 = \Big(\frac{\sum_{i \neq t} e^{y_i}}{\sum_{j=1}^c e^{y_j}}\Big)^2 \tag{108}$$

On the other hand, the all-class Brier score is

$$\|\boldsymbol{y} - \mathbf{1}_t\|_2^2 = \sum_{i \neq t} p_i^2 + (1 - p_t)^2 = \sum_{i \neq t}\Big(\frac{e^{y_i}}{\sum_{j=1}^c e^{y_j}}\Big)^2 + \Big(\frac{\sum_{i \neq t} e^{y_i}}{\sum_{j=1}^c e^{y_j}}\Big)^2 \tag{109}$$

$$= \frac{(\sum_{i \neq t} e^{y_i})^2 + (\sum_{i \neq t} e^{y_i})^2}{(\sum_{j=1}^c e^{y_j})^2} \tag{110}$$

The numerators are $\sum_{i,j \neq t} e^{y_i y_j}$ and $\sum_{i,j \neq t} e^{y_i y_j} + \sum_{i \neq t} e^{2y_i}$. That is, the all-class Brier score increases the weights of the non-target class by 1. The differentiation between them is one of extent rather than essence.

Besides, by Cauchy-Schwartz inequality, we have

$$(1 - p_t)^2 \leq \|\boldsymbol{y} - \mathbf{1}_t\|_2^2 \leq 2(1 - p_t)^2 \tag{111}$$

that tightly bounds the all-class Brier score with the single-class one.

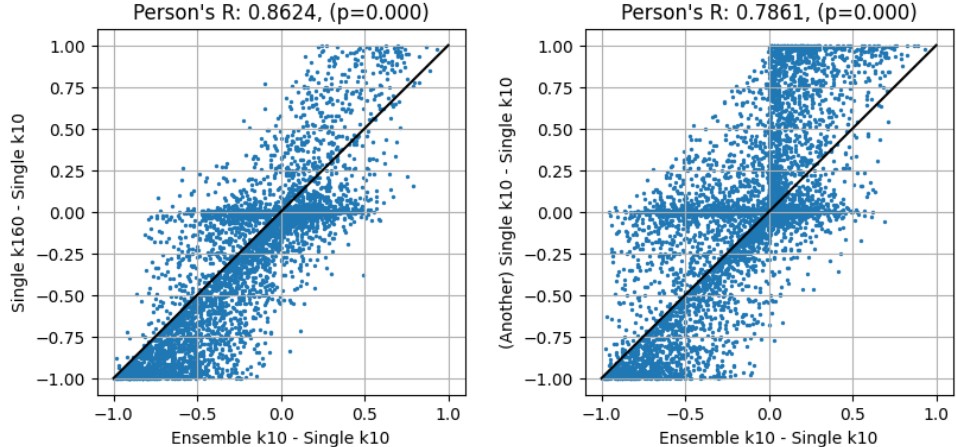

(a) Large Single - Small Single Difference vs. Ensemble - Small Single Difference

(b) Small Single - Small Single Difference vs. Ensemble - Small Single Difference

Figure 8: The comparison between the single model-single model performance difference ($y$-axis) and the ensemble-single model performance difference ($x$-axis) for all testing data. In (1) a large single model is used for the $y$-axis (large single model-small single model). In (2), two independent small single models are used.

## C  ENSEMBLES VS. SINGLE MODELS

Single models and ensembles are compared by comparing the results of

$$\phi(\bar{f}(\boldsymbol{x})) - \phi(f(\boldsymbol{x})) \text{ v.s. } \phi(g(\boldsymbol{x})) - \phi(f(\boldsymbol{x})) \tag{112}$$

where $f$ are small models while $g$ are large models in single-model capacity. Following previous work (Abe et al., 2022b), we implement the results in fig. 8(a). The $y$-axis represents the difference between a large and a small single model. The $x$-axis represents the difference between an ensemble (of small models) and a small single model. Note that since the values represent the Brier score differences, a negative value indicates a performance improvement.

Based on this result, previous work reported that the performance improvement from increasing model capacity and ensembling are very similar. However, we argue that this should be reconsidered carefully. Our analysis demonstrates that these results along with the high Pearson's R values, and almost-zero $p$-values, are all caused by the **distributional equivalence with point-wise distinction** phenomenon revealed in our work. For a $k = 160$ single model $g$ and a $k = 10$ single model $f$, we know that $g$ predicts correctly for a set $U_g \subset \mathcal{X}$ while $f$ predicts correctly for $U_f$. We know that $|U_f| < |U_g|$ almost surely because of the increasing capacity. However, $U_f \subset U_g$ **almost never holds** because of point-wise distinction[2].

**Reasoning.**    The points in fig. 8(a) that have negative $y$ values are those $\boldsymbol{x} \in U_g \backslash U_f$ (predicted correctly by $g$ but wrongly by $f$). Given that these points are misclassified by $f$, due to the distributional equivalence property, they are more likely to be predicted correctly by other $f$s in $\mathcal{F}$, resulting in also negative $x$-values. On the other hand, the points that have negative $y$ values are those $U_f \backslash U_g$ (predicted correctly by $f$ but wrongly by $g$). And because they are already predicted correctly by $f$, due to the distributional equivalence property, they are more likely to be predicted wrongly by other $f$s in $\mathcal{F}$, resulting in also positive $x$-values. These two scenarios lead to the positive correlation between $\phi(\bar{f}(\boldsymbol{x})) - \phi(f(\boldsymbol{x}))$ and $\phi(g(\boldsymbol{x})) - \phi(f(\boldsymbol{x}))$.

---

[2]Note that there are infinitely many large model $g$s such that $U_f \subset U_g$, but compared with $\mathcal{G}$, such $g$s' measure should be extremely small. For a sanity check, we compare $U_f$ for all $M = 100$ CNN models with $k = 10$ and $U_g$ for all $M = 100$ CNN models with $k = 160$. No examined of $(f, g)$ pair satisfies the subset relationship $U_f \subset U_g$.

Here we show a counter-example – The justification above **does not** require $|U_g| > |U_f|$. That is, $g$ does not have to be a larger single model compared with $f$. We can obtain the same positive correlation with two $k = 10$ CNN models. This counterexample is shown in fig. 8(b). For the $y$-axis, instead of using a large single model $g$, we use another small model with $k = 10$. We can observe that the exact same trend holds in both fig. 8(a)(b).

## D  ADDITIONAL FIGURES FOR DISTRIBUTIONAL EQUIVALENCE

In addition to the demonstrative figure in fig. 2, we carry out comprehensive experiments to verify the distributional equivalence property. For model structure, we investigate the direct connection like CNNs and the skip connection like ResNets. We test three datasets, including small models such as CIFAR-10, and CIFAR-100 and a more complex dataset like TinyImagenet. For each dataset and model structure, we scale the single-model capacity through a width parameter $k$ in $\{10, 20, 40, 60, 160\}$. As a result, there are a total of $2 \times 3 \times 5 \times M = 3000$ models. The results for CNNs and ResNets are presented in fig. 9 and fig. 10, respectively. It can be clearly observed that all models satisfy the distributional equivalence property. That is, within each subplot, all $M = 100$ independently trained models have the equivalent distribution of the predictions for the output class. Besides, almost all the mass of the distributions falls at two end points $\{0, 1\}$. This also provides support for the neural collapse assumptions.

Furthermore, in each row, from left to right are an increase in the data complexity. Since the model capacity remains the same, the prediction level $\rho$ (i.e. $\mathbb{E}_{X \sim p_X}[f(X)]$) decreases drastically. On the other hand, in each column, from top to bottom are an increase in the model capacity, while the dataset stays invariant. It can be observed that for the same $p_X$, $\rho$ increases with the single-model capacity. This naturally leads to a performance improvement under monotonic metrics such as the NLL or the Brier score.

Additional results of point-wise distinctions are shown in figs. 11 and 12. The criteria are identical to the results of fig. 2(b). Only the first five models (seed from 1 to 5) are plotted for better visualization. It can be clearly observed that models differ significantly regarding each input. Combined with figs. 9 and 10, the distributional equivalence property is comprehensively verified.

## E  ADDITIONAL FIGURES EMPIRICAL RESULTS

Here we show additional experiments as complementary to the manuscript. fig. 14 shows additional results to fig. 4 on ResNet instead of CNNs. In the manuscript, figs. 6 and 7 show the results of CNNs on CIFAR-10 and CIFAR-100. Here we present other results such as CNNs on all datasets in figs. 15 and 17. And the results of ResNets on all datasets are presented in figs. 16 and 18. In other words, figs. 15 and 16 are the additional results to fig. 6 and figs. 17 and 18 are additional results to fig. 7.

We also plot the distributions of the testing accuracy of all models in fig. 13. he top row shows the results of CNNs while the bottom row shows the results of ResNets. Each color represents the distribution from models of identical width. It's observed that when the models only differ in the training stochasticity from SGD, their testing performances are very similar.

Besides, to further validate the discovered distributional equivalence property, we include models trained using schemes other than the standard SGD. Here we test SGD with momentum. We set the learning rate as lr=1e-3, and momentum as 0.9. For another variant, we further include a weight decay at 5e-4. The results are shown in fig. 19. Here we use CNN models with $k = 20$ and CIFAR-10 dataset.

We also present the error rates of all the single models and ensembles in fig. 20 as a complementary result to figs. 6, 15 and 16.

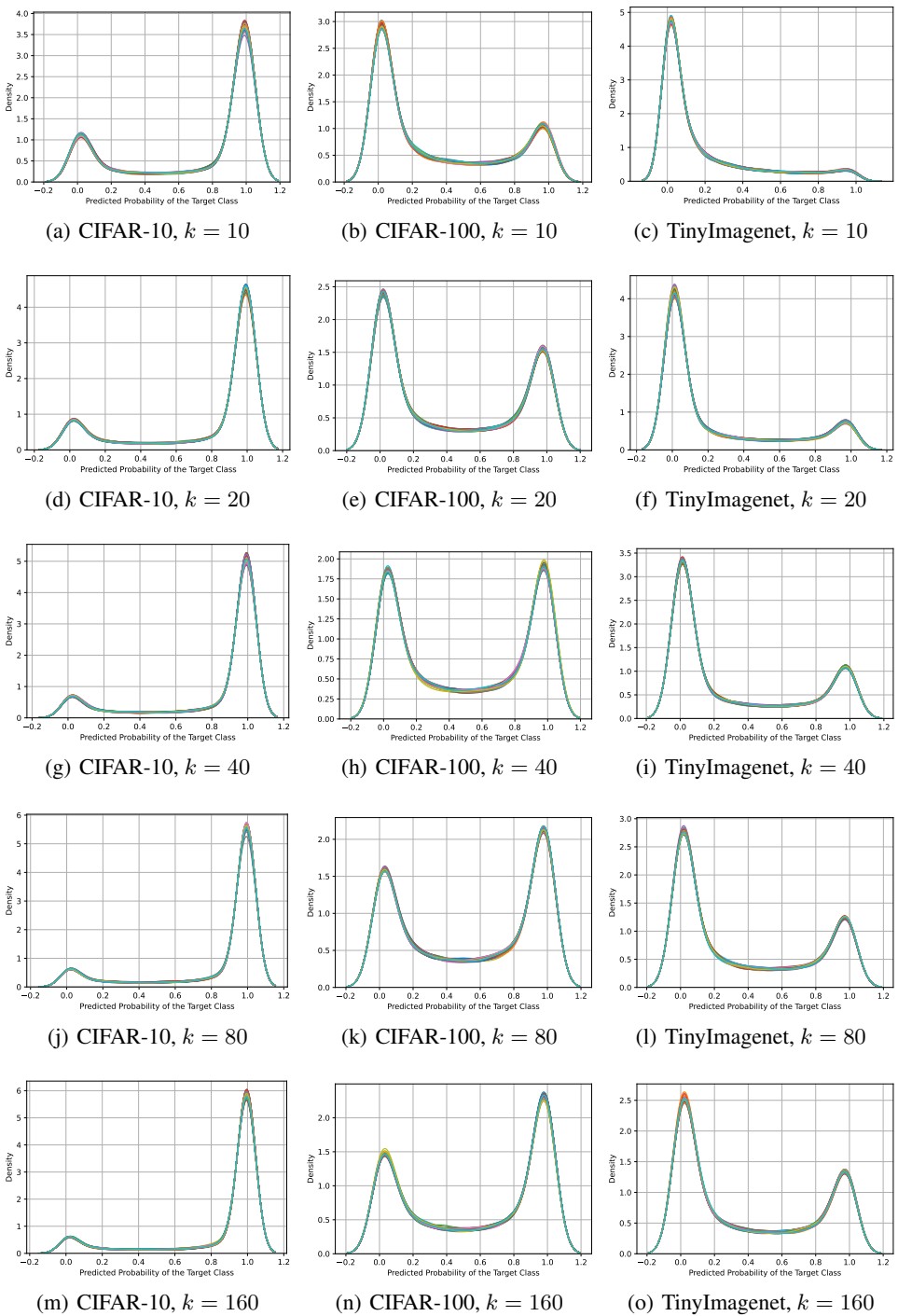

Figure 9: The demonstration of the KDE of the prediction of the target class. The density is estimated using all testing samples. Here CNN models with different single-model capacities (widths) and three datasets are tested. For each figure, all $M = 100$ models are plotted.

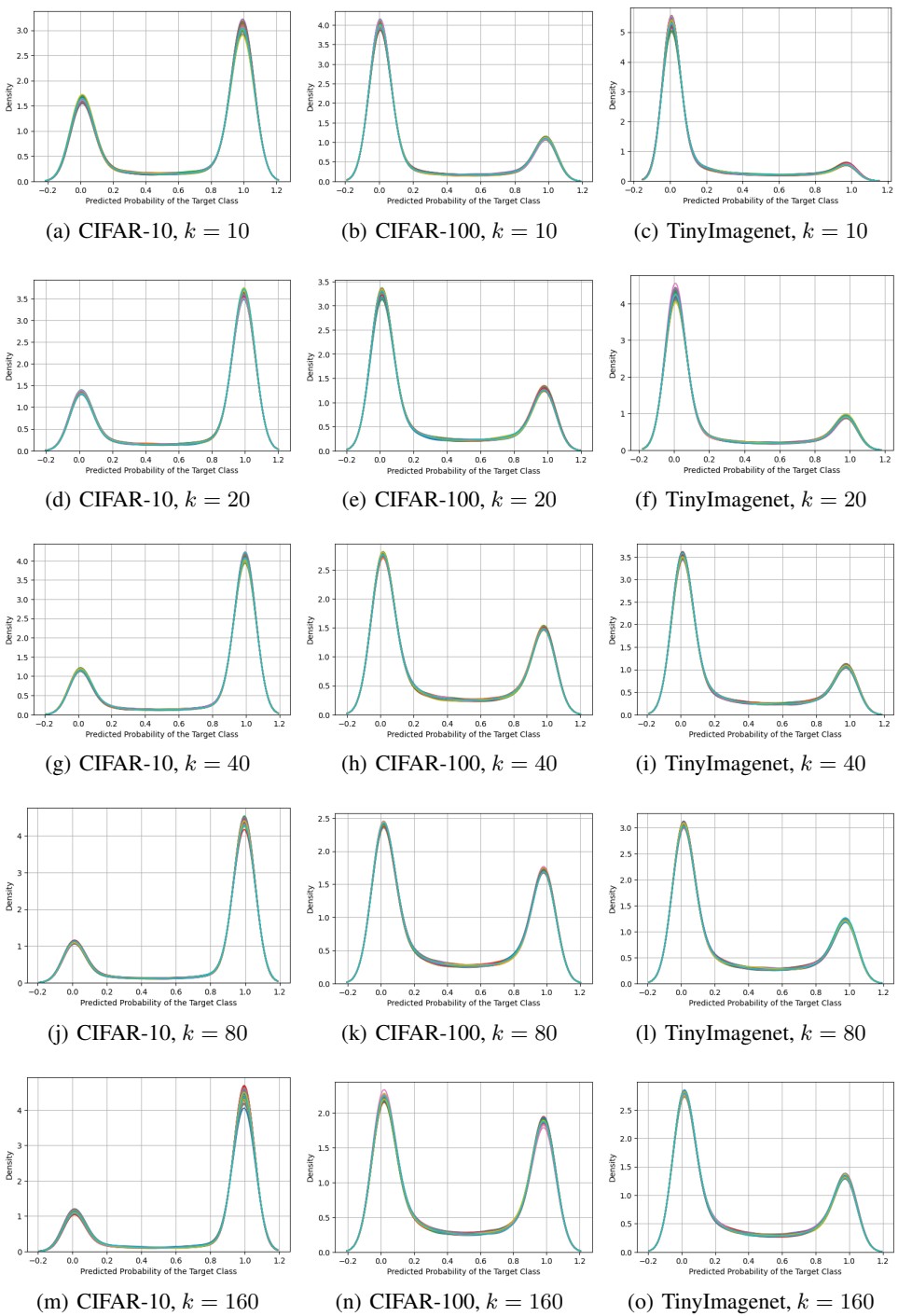

Figure 10: The demonstration of the KDE of the prediction of the target class. The density is estimated using all testing samples. Here ResNet models with different single-model capacities (widths) and three datasets are tested. For each figure, all $M = 100$ models are plotted.

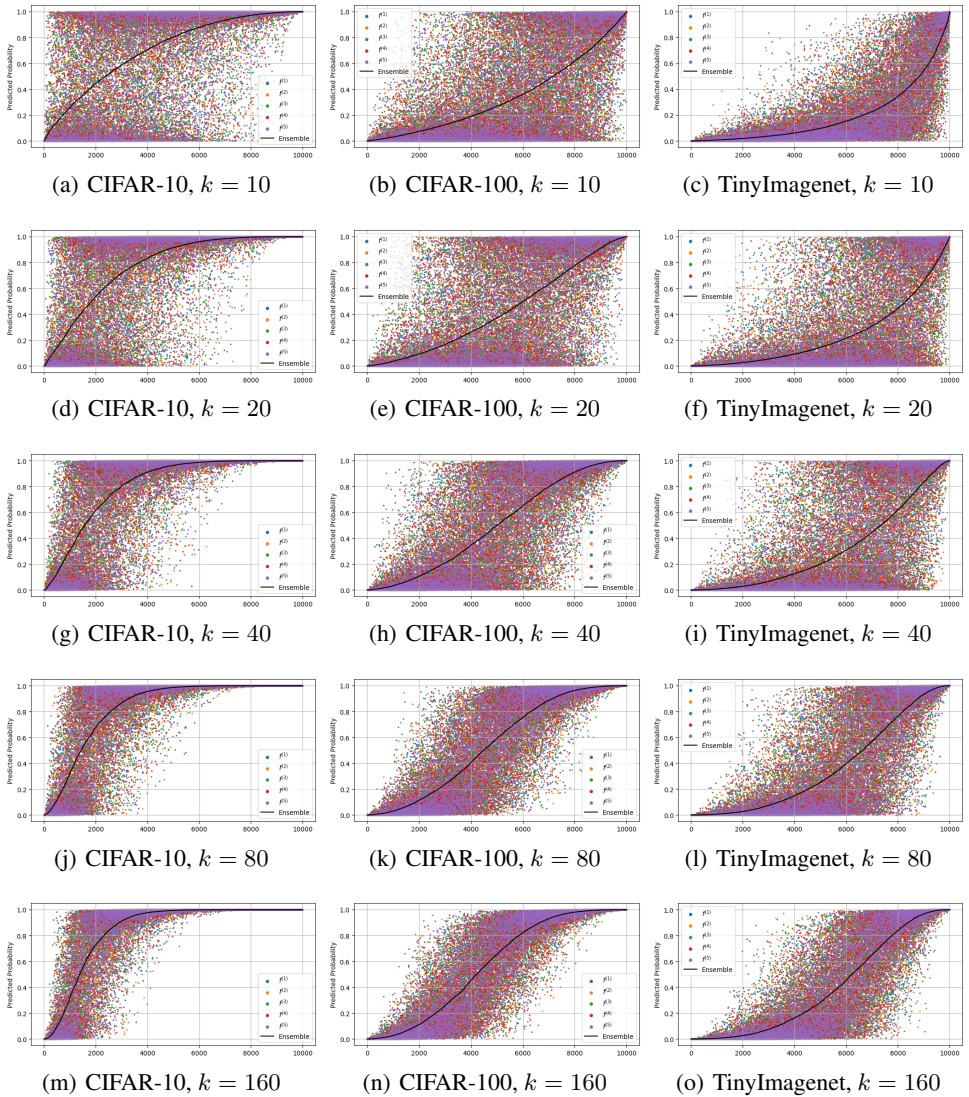

Figure 11: The point-wise prediction of the target class. Here CNN models with different single-model capacities (widths) and three datasets are tested. For each figure, we plot the first 5 models (*seed* from 1 to 5). Each point represents a specific prediction $f^{(i)}(x_j)$. It is demonstrated clearly that although models tend to agree on certain samples, the point-wise predictions vary significantly.

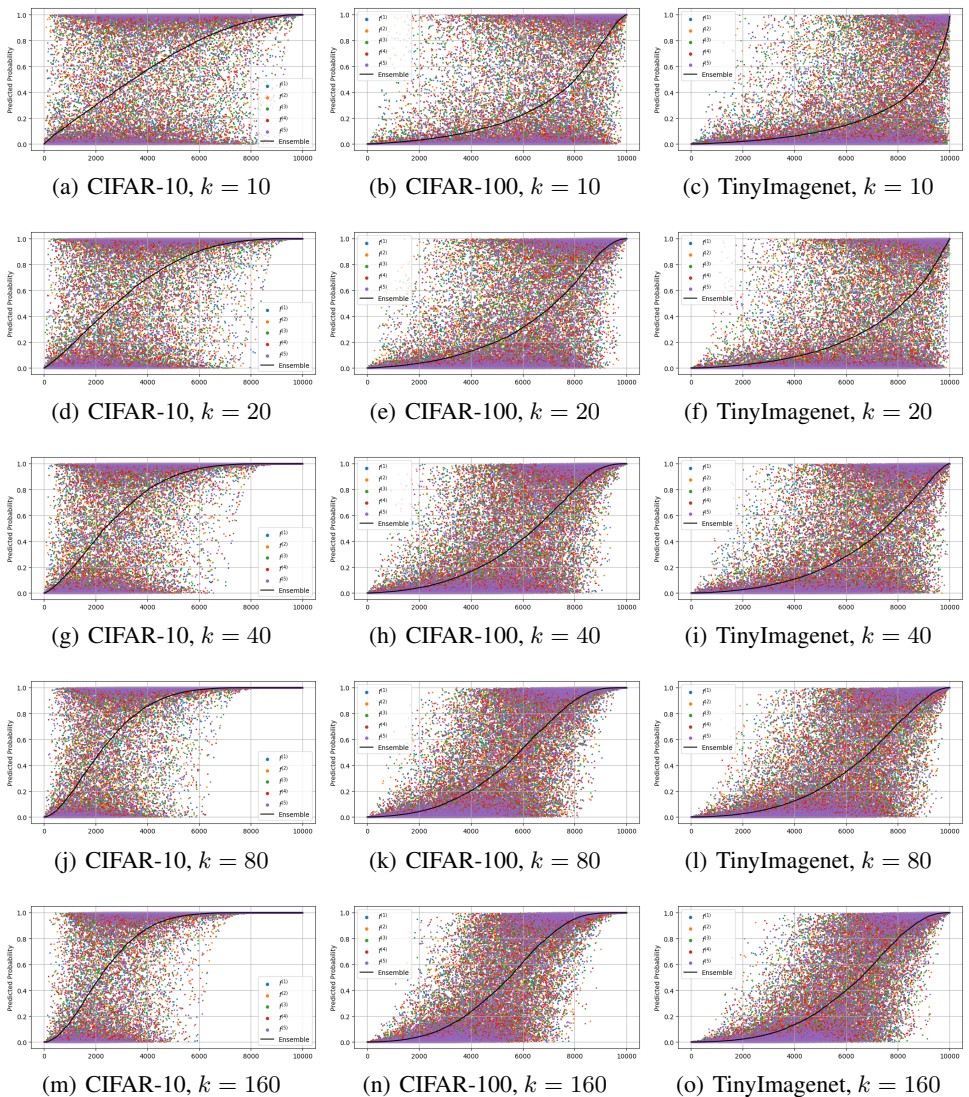

Figure 12: The point-wise predictions of the target class. All testing samples are presented. Here ResNet models with different single-model capacities (widths) and three datasets are tested. For each figure, we plot the first 5 models (*seed* from 1 to 5). Each point represents a specific prediction $f^{(i)}(x_j)$. It is demonstrated clearly that although models tend to agree on certain samples, the point-wise predictions vary significantly.

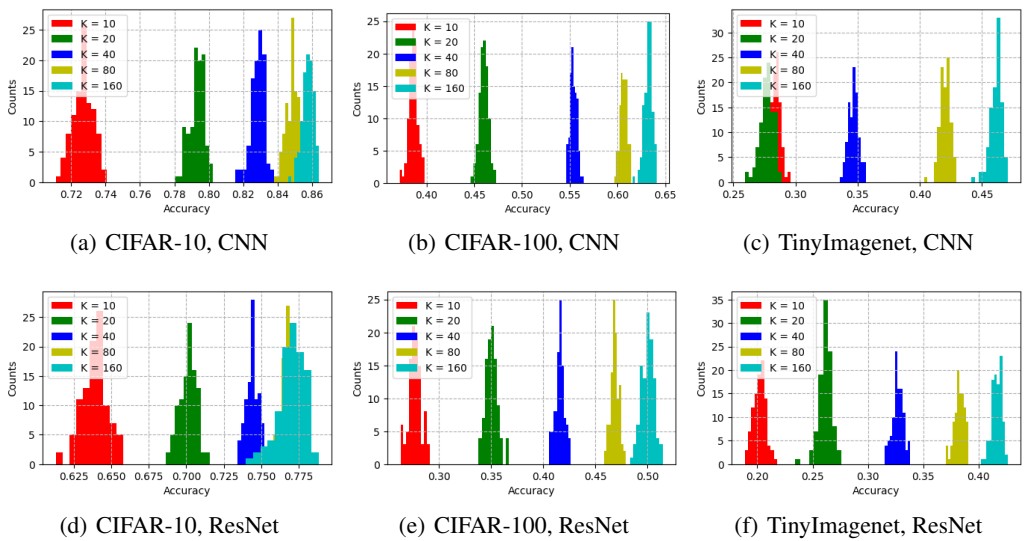

Figure 13: The illustration of the distribution of the testing accuracy of all single models. The top row shows the results of CNNs while the bottom row shows the results of ResNets. Each color represents the distribution from models of identical width.

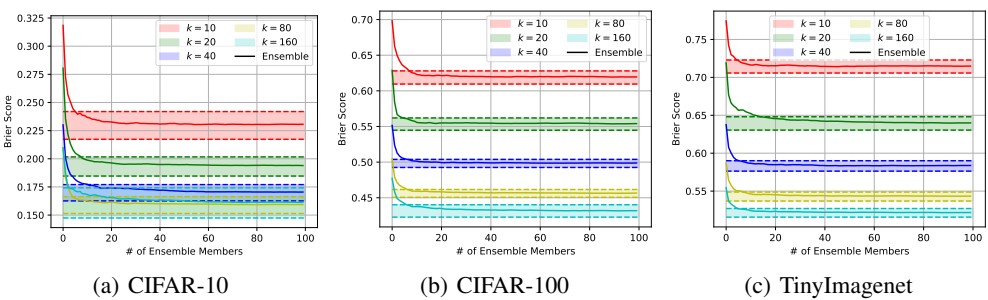

Figure 14: The estimation (regions) and the empirical results (solid curves) of the Brier score of original models. The $x$-axis represents the number of ensemble members (a.k.a. ensemble capacity), and the $y$-axis represents the Brier score of the ensemble. Dashed lines represent the standard deviations. The tested model is ResNet.

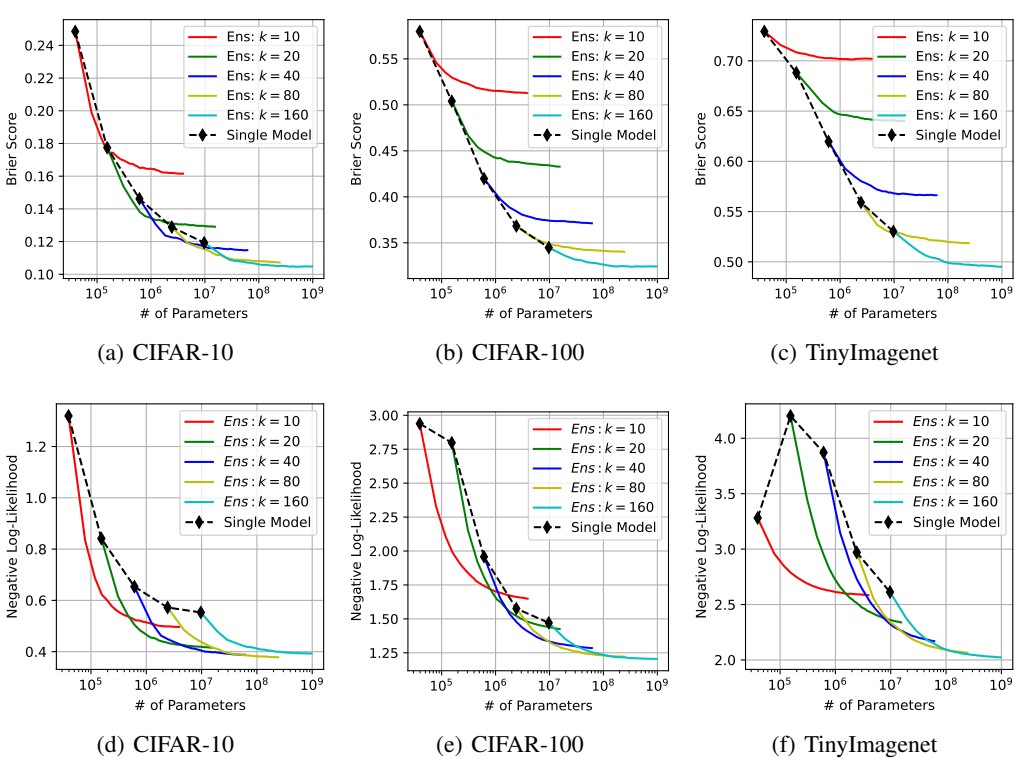

Figure 15: The performance comparison between the scaling of a single model (black dashed curves) and increasing the number of ensemble members (colorful solid curves). In each ensemble (i.e. each colorful curve), $M$ varies from 1 to 100. The results are generated using CNN models.

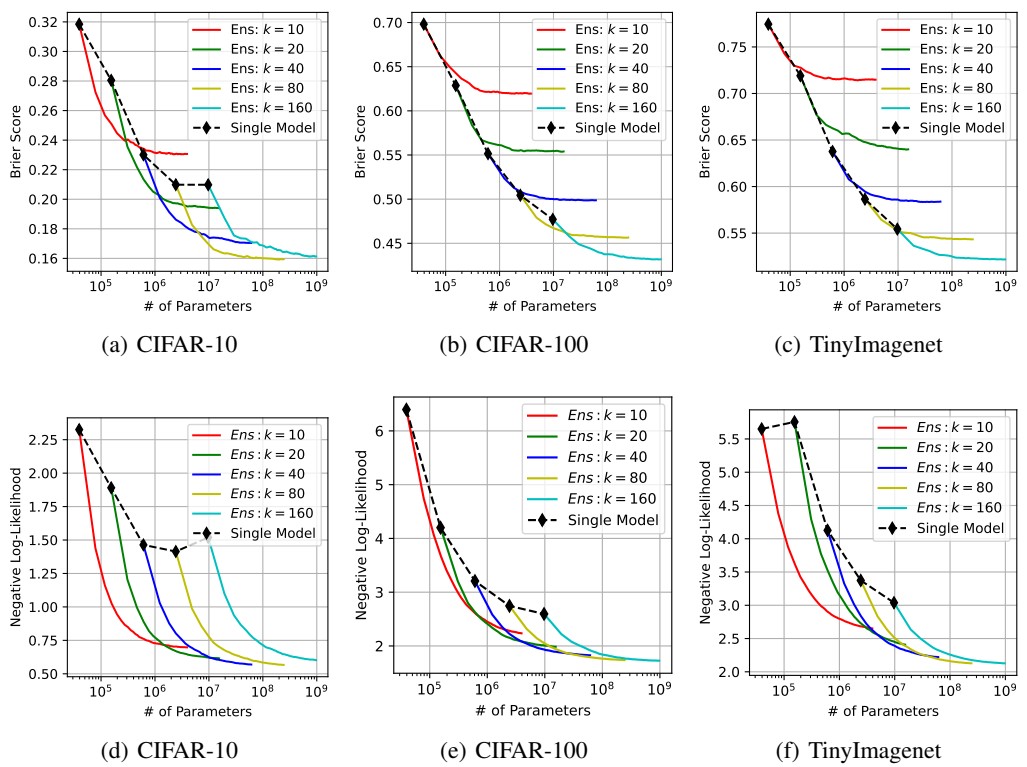

Figure 16: The performance comparison between the scaling of a single model (black dashed curves) and increasing the number of ensemble members (colorful solid curves). In each ensemble (i.e. each colorful curve), $M$ varies from 1 to 100. The results are generated using ResNet models.

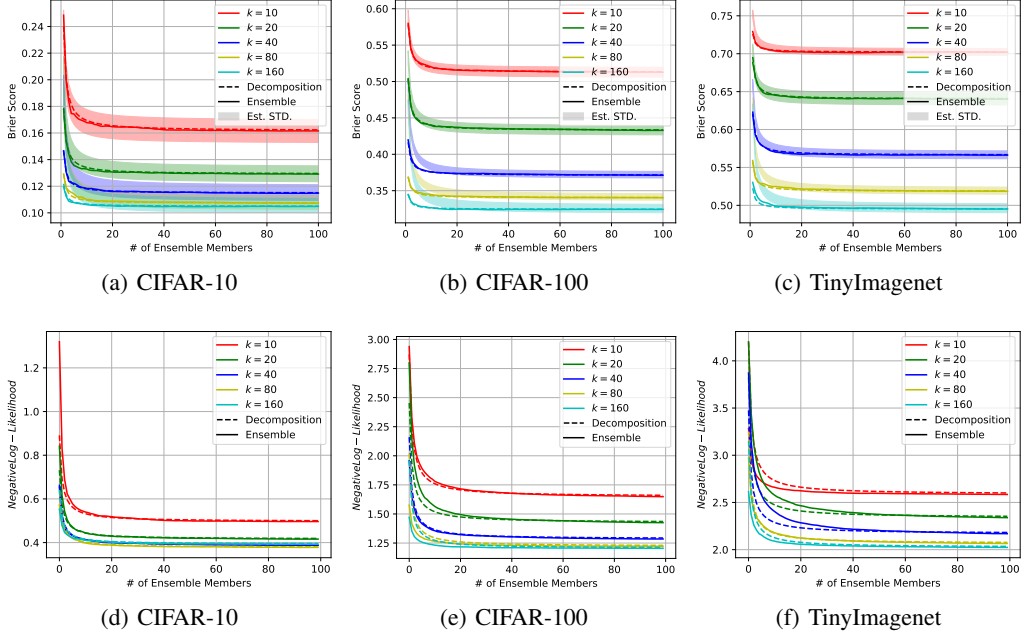

Figure 17: The verification of theorem 5.1. The theoretical results (dashed curves) are compared with the empirical results of ensembles using $M$ models for (a)(b)(c) the Brier score and (d)(e)(f) the negative log-likelihood. The ensemble curves are computed using $p_F$. Results are generated using CNNs.

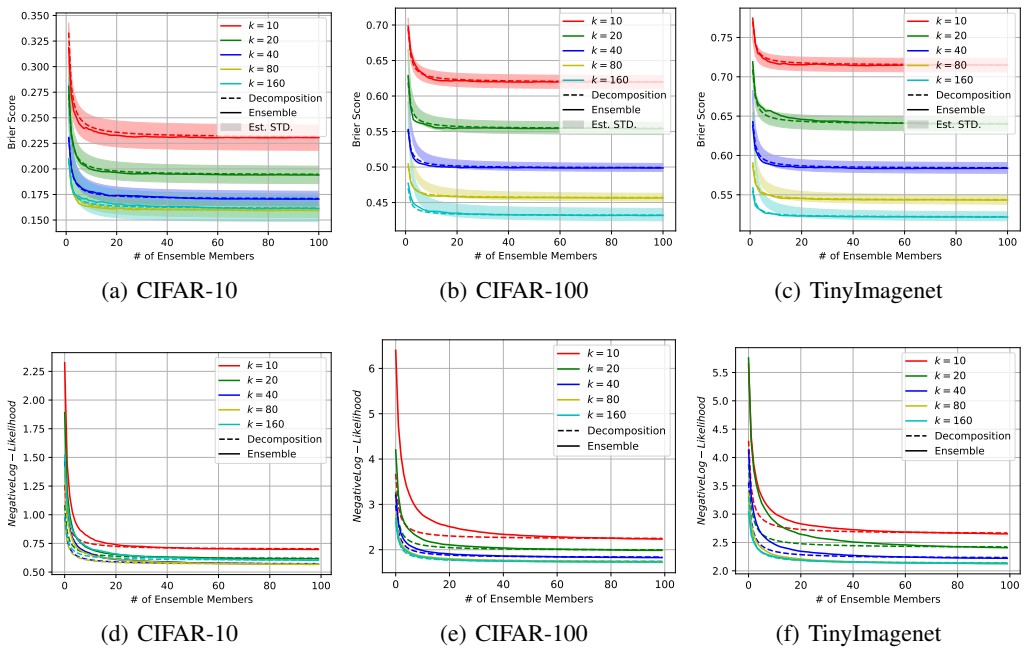

(a) CIFAR-10      (b) CIFAR-100      (c) TinyImagenet

(d) CIFAR-10      (e) CIFAR-100      (f) TinyImagenet

Figure 18: The verification of theorem 5.1. The theoretical results (dashed curves) are compared with the empirical results of ensembles using $M$ models for (a)(b)(c) the Brier score and (d)(e)(f) the negative log-likelihood. The ensemble curves are computed using $p_F$. Results are generated using ResNets.

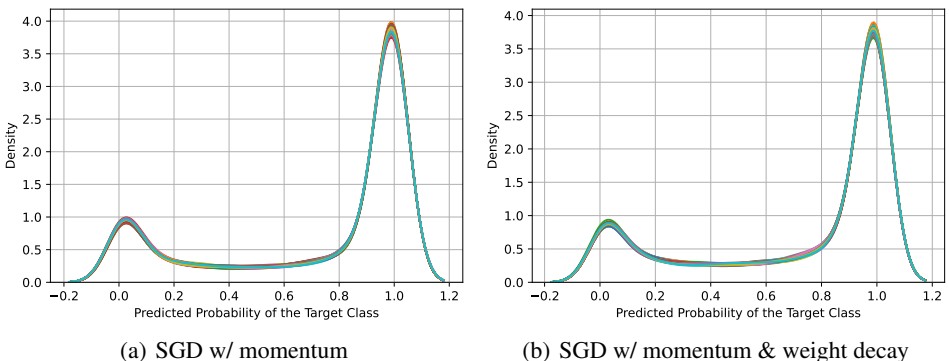

(a) SGD w/ momentum      (b) SGD w/ momentum & weight decay

Figure 19: The demonstration of the KDE of the prediction of the target class. The density is estimated using all testing samples. For each figure, all $M = 100$ models are plotted. The models are CNNs trained with (1) momentum and (2) momentum & weight decay. All testing samples are presented.

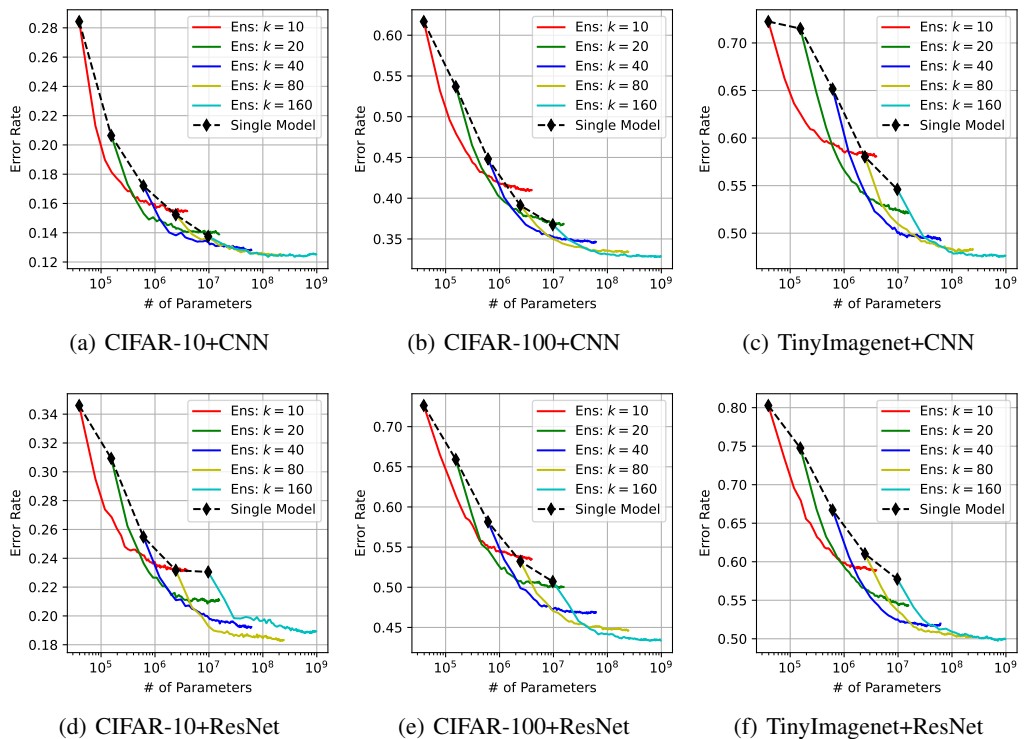

Figure 20: The performance comparison between the scaling of a single model (black dashed curves) and increasing the number of ensemble members (colorful solid curves). In each ensemble (i.e. each colorful curve), $M$ varies from 1 to 100. The results are generated using CNN models.

