# OpenReview forum: "Agree to Disagree: Demystifying Homogeneous Deep Ensembles through Distributional Equivalence"
_ICLR.cc/2025/Conference — ICLR 2025 Poster_

### Official Review · Reviewer_CxeF · 2024-11-01

**Soundness:** 3
**Presentation:** 3
**Contribution:** 4
**Rating:** 8
**Confidence:** 3

**Summary:**

This paper examines why deep ensembles are effective by proposing distributional equivalence, the idea that ensemble members produce predictions with identical distributions. The authors argue this property is key to the performance gains of deep ensembles, offering a new explanation beyond the Jensen gap. They validate this theory through experiments and introduce an estimation scheme that allows ensemble performance to be predicted using just two models, reducing evaluation costs and highlighting the unique benefits of deep ensembles over scaling single models.

**Strengths:**

1. This paper introduces a fresh theoretical perspective on deep ensembles by proposing the concept of distributional equivalence among ensemble members. This approach offers an alternative to the usual Jensen gap explanation, providing insight into why deep ensembles work by analyzing homogeneous model distributions.
2. The proposed method of estimating ensemble performance using only two models is a practical contribution that could streamline the application of ensembles in resource-constrained settings.
3. The authors have conducted comprehensive experiments across multiple datasets and architectures, providing strong empirical support for their theoretical claims.

**Weaknesses:**

1. The theoretical framework relies on assumptions about neural collapse, which may not apply universally across various datasets and model configurations. This reliance could limit the generalizability of the findings, especially in cases with limited data or high variability.
2.Deep ensembles are frequently applied to enhance uncertainty estimation, the paper mainly emphasizes performance improvements, leaving the effect of distributional equivalence on uncertainty quantification undiscussed.

**Questions:**

1. The experiments have included the CIFAR-10 and TinyImageNet, which may not represent the complexity of real-world data, such as text data or multi-modal data. Could the authors discuss how dataset complexity might affect the distributional equivalence property, especially for datasets with higher-dimensional or more variable features?
2. How sensitive is the distributional equivalence property to various training hyperparameters, such as learning rate or initialization seed? An analysis of this could help in understanding the robustness of it across different training configurations.
3. The paper introduces a method for performance estimation using two models. Could the authors comment on the potential for scaling this approach for ensembles of significantly larger capacity.

---

> ### Author Response · Authors · 2024-11-21
> **Response to Reviewer CxeF**
>
> We thank the reviewer very much for acknowledging the theoretical contributions and the comprehensive experiments. We now answer the reviewer's questions as follows.
>
> **Q1. The Neural Collapse Assumption**
>
> In this work, we are focusing on interpolation models that fit the training set. This is supported by the recent studies on the terminal phase of training DNNs, where overfitting does not harm the models' performance (e.g. [2,4,5]). And this has become a common practice in training models at scale. Therefore, it is safe to assume that the models fit the training distribution completely and have become confident.
>
> **Q2. Uncertainty Estimation**
>
> We appreciate the reviewer for mentioning the application of uncertainty estimation of deep ensembles. In fact, our theoretical analysis of deep ensembles also sheds light on the uncertainty estimation of deep ensembles. We now give a brief explanation of this benefit and will add a more detailed analysis in the updated manuscript:
>
> Previous work [1] has demonstrated that for a single input $\mathbf{x}$, the ensemble uncertainty can be decomposed as the summation of the ensemble diversity and the average single model uncertainty:
> $$U(\bar{f}(\mathbf{x}))=Var_{p_F}[F(\mathbf{x})]+\mathbb{E}_{p_F}[U(F(\mathbf{x}))]$$
>
> First, for the ensemble diversity $Var_{p_F}[F(\mathbf{x})]$ itself, our results section 4.2 demonstrate that it should be understood with the actual distributions of the predictions. Eq 6, 7 (L355-366) further shows that the point-wise diversity is highly associated with the ensemble predictions. That is, the more confident the ensemble is at this data $\mathbf{x}$, the lower the ensemble diversity will be. Previous works fall short in this analysis as the differences among $\mathbb{x}$ are not correctly assessed.
>
> For instance, [1] test the hypothesis that the ensemble diversity $Var_{p_F}[F(\mathbf{x})]$ is responsible for improved UQ and conclude that if the ensemble diversity is responsible, then it is suggested that "the performance gains from ensembling are somehow fundamentally different than the performance gains from increasing a single model’s capacity". Since the experiments in [1] also suggest that the performance gains are similar, it was then concluded that the ensemble diversity is not responsible for the improved UQ.
> Our results show a deeper understanding of this issue. From Theorem 5.1, it is clear that the ensemble capacity $M$ and the single model capacity $k$ affect the performance gain in completely different ways. As for the observed "similar performance gain" in Fig. 2 of [1], we add a discussion section in Appendix C of the updated manuscript to show that the conclusion was insufficiently justified.
>
> **W1. Experiments Scale**
>
> We thank the reviewer for the question. We answer this question in the two following aspects:
>
> - First, we deduce that the revealed distributional equivalence property shall hold in more complicated dataset such as Imagenet. This is because the phenomenon of neural collapse has been observed for an even more complicated dataset. Besides, the consistency from CIFAR-10 to CIFAR-100 and to TinyImagenet shows a good generalization of the phenomenon. As a result, it is deduced that for even more complicated datasets, the trained models will have lower $\rho$ (e.g. see each row in Fig 8 and 9). That is, the scaling of models and datasets should only influence the resulting model performance $\rho$ instead of the neural collapse phenomenon.
>
> - Second, we would like to emphasize that the experiment scale in our work is the greatest in the existing work regarding this topic, surpassing previous work by a large margin. We follow the training criterion of the important work of deep double descent [2], where the single models' capacity is limited to $k=64$. We expand this to $k=160$, resulting in multiple times larger models. Besides, due to the requirement of training multiple models, previous works are mainly carried out on the CIFAR dataset, where we extend this to the much more complicated TinyImagnet dataset.

---

> ### Author Response · Authors · 2024-11-21
>
> **W2. Distributional Equivalence Sensitivity**
>
> The distributional equivalence property is very stable regarding the hyperparameters. Different training setups (e.g. solvers, leaning rates, etc.) can affect the distribution $p_F$ and the ratio $\rho$. For example, if the models are trained using other schemes, the models may achieve higher/lower $\rho$ values. But within each $p_F$, the trained models follow the distributional equivalence property.
>
> In Appendix E, we add the results of models trained using momentum and momentum & weight decay. In the final version, we will add more experiments regarding other hyperparameters, such as learning rates.
>
>
> **W3. Performance Estimations with Large Capacity**
>
> We thank the reviewer for bringing this up. We would like to clarify that our proposed theoretical results are capable of handling any level of capacity.
> As derived in Theorem 5.1, the expected loss of the ensemble of $M$ individual models is determined by two factors. The first one is the expected loss of the population mean of the entire distribution $p_F$, and the second term is positive, divided by the number of ensemble members $M$. As a result, this framework suggests that the loss decreases inverse-proportionally as $M$ increases. When the ensemble capacity $M$ is sufficiently large, the loss converges to the loss of the population mean of the entire distribution. These theoretical results are all verified empirically in Fig. 7.
>
>
> **References**
>
> [1] Taiga Abe, Estefany Kelly Buchanan, Geoff Pleiss, Richard Zemel, and John P Cunningham. Deep ensembles work, but are they necessary? Advances in Neural Information Processing Systems, 35: 33646–33660, 2022b.
>
> [2] Preetum Nakkiran, Gal Kaplun, Yamini Bansal, Tristan Yang, Boaz Barak, and Ilya Sutskever. Deep double descent: Where bigger models and more data hurt. Journal of Statistical Mechanics: Theory and Experiment, 2021(12):124003, 2021.
>
> [3] Kobayashi, Seijin, Johannes von Oswald, and Benjamin F. Grewe. "On the reversed bias-variance tradeoff in deep ensembles."
>
> [4] Vardan Papyan, XY Han, and David L Donoho. Prevalence of neural collapse during the terminal phase of deep learning training. Proceedings of the National Academy of Sciences, 117(40): 24652–24663, 2020
>
> [5] Cao, Y., Chen, Z., Belkin, M., and Gu, Q. (2022). Benign overfitting in two-layer convolutional neural networks. Advances in neural information processing systems, 35:25237–25250.

---

> > ### Comment · Reviewer_CxeF · 2024-11-25
> >
> > I want to thank the authors' rebuttal, which addressed my concerns. I have increased my confidence.

---

### Official Review · Reviewer_eaji · 2024-11-02

**Soundness:** 3
**Presentation:** 4
**Contribution:** 3
**Rating:** 8
**Confidence:** 4

**Summary:**

Previous theory of Deep Ensemble owing the benefit of ensembling to the diversity of ensemble members with "trivial" Jasen Gap of the losses without considering the data distribution. Instead, this paper introduces the distributional equivalence of the trained members from global data distribution, which further explains Deep Ensemble works with the equivalent output distribution globally but with distinct sample-wise predictions for a group of samples. Under this perspective, the paper further provides the theory of how sample-wise diversity and the number of members contribute to the performance. Besides, it provides a method to predict the ensemble performance with only two models.

**Strengths:**

1. The paper is easy to follow and well-organized.
2. The introduced "distributional equivalence" that all members produce similar prediction distribution across a group of samples is novel, and has been neglected by previous works.
3. The theory is technically sound and the prediction of ensemble performance with only two models is also interesting.

**Weaknesses:**

1. What does the "only differ in the training stochasticity" actually mean? i.e., where does stochasticity come from? from the shuffled dataset? or random initialization?
2. An interesting question is why the homogeneous members demonstrate the "distributional equivalence" property. Does it come from the fixed dataset bias, fixed architecture, etc.?

**Questions:**

See Weaknesses.

---

> ### Author Response · Authors · 2024-11-21
> **Reponse to Reviewer eaji**
>
> We thank the reviewer very much for the acknowledgment of the contribution of our work. We address the remaining concerns of the reviewer as follows.
>
> **W1. Training Stochasticity**
>
> The reviewer's understanding of the term "training stochasticity" is correct. Here we leverage the stochastic gradient descent (SGD) algorithms as suggested in various previous works [1-4]. In SGD, the stochasticity rises from both the random initialization and the random order of the batches of data.
>
> **W2. Sources of Distributional Equivalence**
>
> We appreciate the reviewer for the insightful question. We would like to consider the distributional equivalence property as a deeper understanding of the influence of model capacity. It has been commonly agreed that when the model capacity (e.g. ResNet-18) and the training scheme (e.g. SGD) are fixed, the resulting models would have "similar performance" (e.g. testing accuracy, testing loss, etc.). However, relations between the resulting performance and the capacity/scheme are unclear and difficult to identify. Distributional equivalence, on the other hand, pushes this "similar performance" one step ahead and reveals that not only do the models have similar accuracy, but they also have very similar distributions of the outputs. That is, when the capacity is fixed, if a model generalizes to a specific part of the testing distribution, another part of the testing distribution is destined to be missed continuously.
> This could be attributed to the counterintuitive geometric properties of high-dimensional data manifold, where most volumes are distributed near the surface.
> However, we admit that this is an interesting but very complicated phenomenon. Thus we leave the detailed study to future work.
>
>
> **References**
>
> [1] Preetum Nakkiran, Gal Kaplun, Yamini Bansal, Tristan Yang, Boaz Barak, and Ilya Sutskever. Deep double descent: Where bigger models and more data hurt. Journal of Statistical Mechanics: Theory and Experiment, 2021(12):124003, 2021.
>
> [2] Ekaterina Lobacheva, Nadezhda Chirkova, Maxim Kodryan, and Dmitry P Vetrov. On power laws in deep ensembles. Advances In Neural Information Processing Systems, 33:2375–2385, 2020.
>
> [3] Balaji Lakshminarayanan, Alexander Pritzel, and Charles Blundell. Simple and scalable predictive uncertainty estimation using deep ensembles. Advances in neural information processing systems, 30, 2017
>
> [4] Huang, Gao, Yixuan Li, Geoff Pleiss, Zhuang Liu, John E. Hopcroft, and Kilian Q. Weinberger. "Snapshot Ensembles: Train 1, Get M for Free." In International Conference on Learning Representations. 2022.

---

> > ### Comment · Reviewer_eaji · 2024-11-25
> >
> > Thanks to the rebuttal efforts of the authors, I will keep my rate.

---

### Official Review · Reviewer_C32A · 2024-11-03

**Soundness:** 2
**Presentation:** 4
**Contribution:** 2
**Rating:** 6
**Confidence:** 3

**Summary:**

The paper discovers that members of homogeneous ensembles have the distributional equivalence property. Based on this property, the paper answers the fundamental question that why deep ensembles work. Additionally, a way to estimate the performance of deep ensembles is proposed. The paper further discusses the bias-variance tradeoff in deep ensembles and compares the effectiveness of scaling a single model versus scaling deep ensembles. Comprehensive theoretical and empirical analyses are provided to support the claims in the paper.

**Strengths:**

1. Theoretical findings are fruitful and solid.
2. Insightful discussions about related works can effectively engage readers.

**Weaknesses:**

1. In Line 135 of the paper, aurthors mention that the unsatisfactory results of the majority vote classifier inspire them to explore overlooked conditions, which turns out to be the distributional equivalence property. Then, why is there no further effort to explain the majority vote classifier after finding the overlooked condition?
2. The distributional equivalence property is observed on CIFAR-10 dataset. However, on such small datasets, it's known that the trained DNNs tend to be quite overconfident, especially in the case where weak or even no regularization is adopted. Combined with the fact that DNNs are highly accurate on small datasets, the observed distributional equivalence is not surprising at all. To justify the distributional
equivalence property in general cases, larger datasets and/or stronger regularization settings need to be considered.
3. In Section 4.4, results are mainly derived for the Brier score, while those for NLL are missing.
4. In Fig. 6, it's better to include plots for error rate in additon to losses.
5. In Line 676, authors note that SGD is used without momentum or data augmentation. However, these techiques are crucial for DNNs to reach high performance. If they are not considered, the results of the paper could have limited practicality.

**Questions:**

Please see the weakness part.

---

> ### Author Response · Authors · 2024-11-21
> **Reponse to Reviewer C32A**
>
> We appreciate the reviewer very much for the acknowledgment of our work.
>
> **W1. Majority Vote**
>
> Although the majority vote is also a way of combining predictions of independent models, it is a relatively less focused way of ensembling in the studies of deep ensembles. Specifically, the deep ensembles are introduced and defined as the arithmetic average of the models (e.g. [1-3]). As a result, we focus mainly on deep ensembles.
> As for the reference to the majority vote in L135, it is meant to emphasize that when exploring the mechanism of the models and ensembles, the trained models have very special mathematical properties and should **not** be considered as general functions.
>
> **W2. Dataset Scales and Distributional Equivalence**
>
> First, we would like to argue that even for CIFAR-10, the distributional equivalence property is not trivial. It might have been established that independent models with the same capacity might have similar performance over the testing set (e.g. accuracy, loss), that is $\mathbb{E}\_{\mathbf{x}\sim p_X}[f(\mathbf{x})]\approx \mathbb{E}\_{\mathbf{\mathbf{x}}\sim p_X}[g(\mathbf{x})]$ for two independent models. However, the property of the distribution of the predictions has never been recognized. The distributional equivalence property suggests a much stronger property on the distribution as
> $$P(f(\mathbf{x})=v)\approx P(g(\mathbf{x})=v)$$
> for any $v$
>
> Second, we clarify that we are focusing on interpolation models that fit the training set. This is supported by the recent studies on the terminal phase of training DNNs, where overfitting does not harm the models' performance (e.g. [5-7]). Therefore, it is safe to assume that the models fit the training distribution completely and have become confident.
>
> Finally, we emphasize that due to the requirement of training numerous models, the scale of the experiments in our work is already the greatest in the existing work regarding this topic, surpassing previous work by a large margin. We follow the training criterion of the important work of deep double descent [2], where the single models' capacity is limited to $k=64$. We expand this to $k=160$, resulting in multiple times larger models. Besides, due to the requirement of training multiple models, previous works are mainly carried out on the CIFAR-10 dataset, where we extend this to the much more complicated TinyImagnet dataset. And compared with studies of deep ensembles, we have $M=100$ ensemble members for each setting, which is the largest to our best knowledge.

---

> ### Author Response · Authors · 2024-11-21
>
> **W3. Results of NLL**
>
> Among the convex and decreasing metrics, Brier score takes the square of the prediction errors, which is compatible with the expectations and variances. As a result, Brier score has been preferred in the analysis of ensembles. For example, the latest existing work on understanding deep ensembles is entirely based on Brier score [1]. Therefore, we leverage this and start from Brier score at the beginning, too. The close relation between Brier score and the variance provides an insightful understanding of the ensemble performances in section 4. Afterward, in section 5, leveraging the Taylor expansion, we are the first to theoretically derive rules regarding the NLL of ensembles. The derived results are also empirically verified in Fig. 7.
>
>
> **W4. Error Rate in Fig. 6**
>
> We appreciate the reviewer very much for the suggestion! We have included the figures of the error rates corresponding to the figures in Fig. 6. The error rate results are shown in Appendix E. It can be clearly observed that the error rates are consistent with the losses.
>
>
> **W5. Training Strategies**
>
> We thank the reviewer very much for pointing this out. We clarify that the training strategy follows the settings in [5], which provide a consistent and scalable way of training numerous models. To resolve the reviewer's concern, we carry out additional experiments to train models using SGD with momentums and weight decay. The results in Appendix E of the updated manuscript show that the revealed distributional equivalence property holds across various training strategies. We will add a more comprehensive study with a more complete set of variants such as different solvers in the final version.
>
>
> **References**
>
> [1] Taiga Abe, Estefany Kelly Buchanan, Geoff Pleiss, Richard Zemel, and John P Cunningham. Deep ensembles work, but are they necessary? Advances in Neural Information Processing Systems, 35: 33646–33660, 2022b.
>
> [2] Balaji Lakshminarayanan, Alexander Pritzel, and Charles Blundell. Simple and scalable predictive uncertainty estimation using deep ensembles. Advances in neural information processing systems, 30, 2017
>
> [3] Kobayashi, Seijin, Johannes von Oswald, and Benjamin F. Grewe. "On the reversed bias-variance tradeoff in deep ensembles."
>
> [4] Ryan Theisen, Hyunsuk Kim, Yaoqing Yang, Liam Hodgkinson, and Michael W Mahoney. When are ensembles really effective? arXiv preprint arXiv:2305.12313, 2023.
>
> [5] Preetum Nakkiran, Gal Kaplun, Yamini Bansal, Tristan Yang, Boaz Barak, and Ilya Sutskever. Deep double descent: Where bigger models and more data hurt. Journal of Statistical Mechanics: Theory and Experiment, 2021(12):124003, 2021.
>
> [6] Vardan Papyan, XY Han, and David L Donoho. Prevalence of neural collapse during the terminal phase of deep learning training. Proceedings of the National Academy of Sciences, 117(40): 24652–24663, 2020
>
> [7] Cao, Y., Chen, Z., Belkin, M., and Gu, Q. (2022). Benign overfitting in two-layer convolutional neural networks. Advances in neural information processing systems, 35:25237–25250.

---

> > ### Comment · Reviewer_C32A · 2024-11-26
> >
> > I thank the authors for addressing my comments. However, my major concern raised in W2 has not yet been satisfactorily resolved. The assumption of neural collapse/overconfident predictions in both theoretical derivations and experiments significantly limits the generalizability of the findings in this work. Thus, I decide to keep my score, which is already positive.

---

### Official Review · Reviewer_fvho · 2024-11-03

**Soundness:** 3
**Presentation:** 3
**Contribution:** 3
**Rating:** 6
**Confidence:** 3

**Summary:**

This paper challenges the common explanation for deep ensemble success, which attributes their effectiveness to Jensen’s inequality. The authors argue that this explanation is insufficient and propose a new perspective based on the **distributional equivalence property** of trained models. According to this view, although ensemble members may differ in their individual predictions, their output distributions (based on the predicted probability for the target class) are statistically identical across the dataset.

Through both theoretical analysis and empirical evidence, the paper demonstrates how this distributional equivalence property helps explain why deep ensembles consistently outperform single models. Additionally, the authors introduce estimation schemes to predict ensemble performance based on the number of members and model capacity, offering insights into the relationship between ensemble capacity and single-model scaling.

**Strengths:**

- The paper offers practical insights into the relationship between global diversity and ensemble performance improvement, as well as the distinct contributions of ensemble capacity versus single-model scaling. The estimation schemes, which allow practitioners to approximate ensemble performance using just two models, provide a computationally efficient approach that could inform more cost-effective system design.
- Beyond merely acknowledging the diversity of ensemble members, the paper provides an in-depth analysis of both **point-wise** and **global diversity**, emphasizing the latter as a key factor often overlooked in prior research.

**Weaknesses:**

- The main contribution, introducing distributional equivalence, may lack generalizability to other loss functions. Please refer to the question section for more details.
- The distributional equivalence condition, a foundational assumption underlying all theorems, seems overly restrictive. Specifically, in the proof of Theorem B.1, lines 692–695, the authors assert that “for arbitrary $\(\hat{f}, \hat{g} \in \hat{F}, P(\hat{f}(x)) = P(\hat{g}(x))\)$” holds  $\(\forall x \in X\)$, which is a strong assumption that I am still not convinced.

**Questions:**

1. Section 4.1’s findings on distributional equivalence focus on the predicted probability for the target class, without considering the full class probability distribution. Could the authors provide an additional analysis that incorporates more information, such as the difference $\(l = p(y) - p(\hat{y})\)$, where $\(\hat{y}\)$ is the predicted class and $\(p(\hat{y})\)$ is the highest probability across all classes? This could give a broader perspective on the consistency of distributional equivalence.

2. Which loss function was used to train the models in Section 4.1? My hypothesis is that the loss function has a significant influence on these findings. For instance, cross-entropy loss focuses on the probability of the target class, so models trained with this objective might naturally exhibit distributional equivalence on this specific metric. In contrast, a loss function such as label smoothing, which considers probabilities across all classes, might lead to different observations, potentially limiting the generalizability of the results in Section 4.1. Could the authors discuss or provide empirical insights on how these findings might vary with alternative loss functions?

---

> ### Author Response · Authors · 2024-11-21
> **Response to Reviewer fvho**
>
> We appreciate the reviewer very much for acknowledging our in-depth analysis and the contributions to understanding deep ensembles. To resolve the remaining concern of the reviewer, we now address all the questions as follows.
>
>
> **W1. & Q2. Loss Functions**
>
> We understand the reviewer's concern regarding the generalizations of the derived theories regarding different loss functions. We answer these questions from the two perspectives that the reviewer mentioned:
>
> 1. Theoretical Derivations: When the distributional equivalence property and neural collapse hold, the theoretical results should generalize easily to all convex and decreasing loss functions, where Brier scores (i.e. square of MSE) and NLL are the two most important types. In Appendix B.8, we have proved the same conclusions as Theorem 5.1 with a completely generalized setting. Should the audience be curious about any kind of loss functions or distributions, they can instantiate $p_{l,X}, p_X$, and $\phi$ to the instances they prefer.
>
> 2. Empirical Settings: We clarify that all models are trained with cross-entropy loss, which is the most commonly used method. This is because deep ensembles are famous for being simple to deploy by directly taking the average of several independently trained models [1]. Therefore, we focus on this most straightforward case -- interpolation models that fit the training set. This is supported by the recent studies on the terminal phase of training DNNs, where overfitting does not harm the models' performance (e.g. [2-4]). Therefore, it is safe to assume that the models fit the training distribution completely and have become confident.
>
>
> **W2. The Distributional Equivalence Property**
>
> We understand the reviewer's concern where the distributional equivalence property seems to be "too strong" in the proof. However, we emphasize that the missing property is exactly what has hindered the previous understanding of deep ensembles.
>
> We have demonstrated in Fig 2(a), 8, and 9 that such a property is prevalent in trained models with common training criteria. In the proofs, we demonstrate this property is responsible for all the "mysterious" performance improvement of deep ensembles. Furthermore, the theoretical results accurately predict the performance of ensembles with an arbitrary number of members -- a theoretically guaranteed scaling law for ensembles. All of these results rely on the *fact* that independently trained models have this distributional equivalence property.
>
> Besides, as mentioned above, we have demonstrated in Appendix B.8 that the theoretical derivations generalize to arbitrary distribution $p_{l,X},p_X$ and loss function $\phi$. In the manuscript, we instantiate them to the most commonly studied ones. However, they can be instantiated to any instances, too.

---

> ### Author Response · Authors · 2024-11-21
>
> **Q1. Predicted Likelihood Vector**
>
> We clarify that the use of the output of the target class is based on the following reasons: (1) Because of the neural collapse phenomenon, the predicted probability of the target class is either very close to 1 or very close to 0. When it's close to 1, the prediction of all other classes is naturally zeros. And when it's close to 0, there is usually one other dominant class. Therefore, taking the target class only is already sufficient for the analysis. (2) Negative log-likelihood only considers the target class. Therefore, it creates unnecessary inconsistency across the analysis. (3) Due to the softmax layer, the output of the target class already contains information from the prediction of **all** classes [5]. Thus they are almost equivalent. We elaborate on this as follows:
>
> Note that discriminative models do not directly predict the likelihood vector $\mathbf{p}\in[0,1]^c$. Instead, they predict the logits $\mathbf{y}\in\mathbb{R}^c$, and then a softmax activation is applied to $\mathbb{y}$ to obtain the likelihood. Here $c$ is the number of classes and let $t$ denote the ground truth class. Then the target-class Brier score is
> $$(1-p_t)^2=(\frac{\sum_{i\ne t}e^{y_i}}{\sum_{j=1}^ce^{y_j}})^2$$
>
> The all-class Brier score is
> $$\|\|\mathbf{y} - \mathbf{1}\_t\|\|\_2^2 = \sum_{i\ne t}p_i^2 + (1-p_t)^2=\sum_{i\ne t}(\frac{e^{y_i}}{\sum_{j=1}^ce^{y_j}})^2+(\frac{\sum_{i\ne t}e^{y_i}}{\sum_{j=1}^ce^{y_j}})^2=\frac{\sum_{i\ne t}(e^{y_i})^2+(\sum_{i\ne t}e^{y_i})^2}{(\sum_{j=1}^ce^{y_j})^2}$$
> It can be observed that the numerators are $\sum_{i,j\ne t}e^{y_iy_j}$ and $\sum_{i,j\ne t}e^{y_iy_j}+\sum_{i\ne t}e^{2y_i}$, respectively. That is, the all-class Brier score upweights the weights of the non-target class by 1. The differentiation between them is one of extent rather than essence.
>
> Besides, by Cauchy-Schwartz inequality, we have
> $$(1-p_t)^2\le\|\|\mathbf{y} - \mathbf{1}_t\|\|_2^2\le 2(1-p_t)^2$$
> that tightly bounds the all-class Brier score with the single-class one.
>
> **References**
>
> [1] Balaji Lakshminarayanan, Alexander Pritzel, and Charles Blundell. Simple and scalable predictive uncertainty estimation using deep ensembles. Advances in neural information processing systems, 30, 2017
>
> [2] Preetum Nakkiran, Gal Kaplun, Yamini Bansal, Tristan Yang, Boaz Barak, and Ilya Sutskever. Deep double descent: Where bigger models and more data hurt. Journal of Statistical Mechanics: Theory and Experiment, 2021(12):124003, 2021.
>
> [3] Vardan Papyan, XY Han, and David L Donoho. Prevalence of neural collapse during the terminal phase of deep learning training. Proceedings of the National Academy of Sciences, 117(40): 24652–24663, 2020
>
> [4] Cao, Y., Chen, Z., Belkin, M., and Gu, Q. (2022). Benign overfitting in two-layer convolutional neural networks. Advances in neural information processing systems, 35:25237–25250.
>
> [5] Wang, Yipei, and Xiaoqian Wang. "“Why Not Other Classes?”: Towards Class-Contrastive Back-Propagation Explanations." Advances in Neural Information Processing Systems 35 (2022): 9085-9097.

---

> > ### Comment · Reviewer_fvho · 2024-11-24
> > **Thanks authors for the responses**
> >
> > I thank the authors for the effort to address my concerns, most of which have been addressed. While I am still concerned about the generalization of the derived theories regarding different loss functions and the assumption about the neural collapse phenomenon (when a model is overconfident in both correct and wrong predictions, leading to near 1 or 0 in the target class), I think this might be acceptable.
> > Based on that, I am leaning toward supporting the paper and raising the score.
> >
> > Thanks

---

### Official Review · Reviewer_wGn1 · 2024-11-05

**Soundness:** 2
**Presentation:** 2
**Contribution:** 2
**Rating:** 6
**Confidence:** 5

**Summary:**

This paper develops a theory to explain why homogeneous deep ensemble can improve the performance.

**Strengths:**

- The experiments of distributional equivalence property in Sections 4.1 and 4.2 are interesting.

**Weaknesses:**

- The setting and assumptions of the distribution $p_F$ which is crucial are not clear. All mentioned is $F \sim p_F$ is a trained model. I believe that this is important for the solidness of the developed theory. At least, we should have something to connect the min over $\mathcal{F}$ and the member function $F \sim p_F$. Otherwise, the theory is loosing.
- The proof of Proposition 3.1 can not be found.
-  It seems to me that the definition of $\hat{f}(x)$ is not meaningful because it is associated with a function $f \in \mathcal{F}$ and it returns 1 if $f(x)$ is a correct prediction and $0$ otherwise. Therefore, $E_{F\sim \hat{p}_F}[F(X)]$ seems to be the percentage of the number of correct predictions when $F \sim p_F$. What is the meaning of applying $\phi$ to this average. Similar question to $\phi(\hat{f}(x))$ in Eq. (8).
-  I have checked the proof of Lemma B1 and Theorem 4.1. They require $E_{x \sim p_x}[\hat{f}(x)]$ = $E_{x \sim p_x}[\hat{g}(x)]$ for every $\hat{f}, \hat{g} \in \hat{\mathcal{F}}$ or the accuracies are equal for every $f,g \in \mathcal{F}$ which is weird for a general function space $\mathcal{F}$. Certainly, we can partly assume this with $f,g \sim p_F$ which is the distribution of well-trained models. However, the support of $p_F$ is only a subset of $\mathcal{F}$. Therefore, I reckon that the minimization in Theorem 4.1 should be over the support of $p_F$.

**Questions:**

Please address my questions in Weaknesses.

---

> ### Author Response · Authors · 2024-11-21
> **Response to Reviewer wGn1**
>
> We appreciate the reviewer very much for the effort in reviewing our work and providing insightful feedback!
>
> We understand that the reviewer's main concern lies in the definitions of the distributions of trained models, functions of models, etc. Here we address the reviewers's concerns as follows:
>
> **W1. The settings of $p_F$**
>
> - The set of trained models: In this work, we focus on models optimized regarding the training distribution. This is a very common setup in the studies of deep ensembles (e.g. [1]) and related topics (e.g. [2,3]). We denote by $\mathcal{F}$ the set of all these models. Empirically, they can be models that interpolate the training set.
> - The distribution of trained models (L179-181): Deep ensembles train $M$ models independently with the same *stochastic algorithm*. This process leads to the distribution $p_F$ of interpolation models over $\mathcal{F}$. As a consequence, training a model is equivalent to sampling from such a distribution $F\sim p_F$.
> - The relations between $p_F$ and $\mathcal{F}$: In summary, as we defined in L181, $p_F$ is a distribution of interpolation models over $\mathcal{F}$. This also answers the reviewer's question about the support of $p_F$. That is, $supp(p_F)\subset\mathcal{F}$. We add a clear and formal definition of the support of $p_F$ in the updated manuscript for a better presentation.
> - Rigorous Definitions: We would also clarify that the set $\mathcal{F}$ and the distributions $p_F$ are distinguished purposely for a rigorous study. For instance, in eq. 1, we specifically define the ensemble as the expectation over the distribution $p_F$ instead of the expectation over the entire set $\mathcal{F}$ of interpolation models because the ensembles implicitly sample trained models under $p_F$ from $\mathcal{F}$ instead of uniformly. On the other hand, as defined in the "best model" should be the best one over the entire set $\mathcal{F}$ instead of the distribution $p_F$.
>
> **W2. The proof of Proposition 3.1**
>
> We thank the reviewer very much for pointing out this. The fact that the lower gap $LG(\mathbf{x})\$ being non-positive follows naturally the arithmetic laws. In plain language, the expectation is always between the maximum and minimum, thus leading to a greater value than the minimum under a monotonic function. We provide the formal proof below and have added this to the upgraded appendix:
>
> Proof: WLOG, let $\phi$ be monotonically increasing. Then
> $$\min_{f\in\mathcal{F}}\{\phi(f(\mathbf{x}))\}=\phi(\min_{f\in\mathcal{F}}\{f(\mathbf{x})\})$$
> Since $\min_{f\in\mathcal{F}}\{f(\mathbf{x})\}\le\mathbb{E}\_{F\sim p_F}[F(\mathbf{x})]$ and $\phi$ is monotonically increasing, we have
> $$\phi(\min_{f\in\mathcal{F}}\{f(\mathbf{x})\})\le\phi(\mathbb{E}\_{F\sim p_F}[F(\mathbf{x})])$$
> Combining them results in
> $$\min_{f\in\mathcal{F}}\{\phi(f(\mathbf{x}))\}\le\phi(\mathbb{E}\_{F\sim p_F}[F(\mathbf{x})])$$
> Therefore
> $$LG(\mathbf{x})=\min_{f\in\mathcal{F}}\{\phi(f(\mathbf{x}))\}-\phi(\mathbb{E}_{F\sim p_F}[F(\mathbf{x})])\le0$$
> This proves the proposition.$\Box$

---

> ### Author Response · Authors · 2024-11-21
>
> **W3. The definition and importance of $\hat{f}$**
>
> We appreciate the reviewer very much for bringing up the importance of $\hat{f}$. However, we respectfully disagree with the reviewer's point that "$\hat{f}$ is not meaningful". Instead, $\hat{f}$ plays a crucial role in understanding the dynamics of deep ensembles. We elaborate on this as follows:
>
> - The approximation to neural collapse: It has been both empirically and theoretically demonstrated in previous work that trained models enter a terminal phase called neural collapse, where deep models can become overconfident without falling into the traditional variance-bias trade-offs. In such scenarios, the predictions of models become extremely close to 0 and 1 for discriminative tasks. Therefore, $\hat{f}$ can be used as an approximation of $f$ in theoretical analyses.
>
> - The approximation is very accurate: Although $\hat{f}$ approximates $f$ in a seemingly strong assumption, we have demonstrated that this approximation is very accurate.
>   - Qualitatively, we have demonstrated in Fig. 2(a), 8, and 9 that this holds true for all kinds of models and datasets.
>   - Quantitatively, we carry out **ALL** of our experiments using the original functions $f$ instead of the approximations. It has been verified that the theoretical results derived using the approximations accurately predict the behaviors of the original models. This provides support to the accuracy of the estimations.
>   - Theoretically, we have proved rigorously in Appendix B.8 (L950-1012) that all the theoretical proof can be generalized to the original models $f$. The use of the surrogate models $\hat{f}$ is purely for tractability. Besides, it enables all the applications such as preciting the dynamics of ensembles of arbitrarily many members using only 2 members.
>
> - The connection between $\hat{f}$ and the corrections: We are glad that the reviewer notices the tight connection between the surrogate models $\hat{f}$ and the correction rate of individual models. In fact, this contributes to a very interesting implication of this work that the dynamic of the entire ensemble system can be identified with very few factors. As demonstrated in Theorem 4.2 - 5.1, the improvement from ensembles is mainly influenced by (1) the correction rate $\rho$ and (2) the correlation between any two models.
>
> **W4. Notations and Conditions in Theorem 4.1**
>
> - The minimization term in Theorem 4.1: First, as mentioned in the response to W1, we appreciate the reviewer very much for pointing out the support of $p_F$ should be defined as $supp(p_F)\subset\mathcal{F}$. However, we emphasize that we are actually proving an even stronger version of Theorem 1. Because $supp(p_F)\subset\mathcal{F}$, the minimization over the support should be no smaller than the minimization over its superset $\mathcal{F}$. Formally:
> $$\min_{f\in \mathcal{F}}\{\mathbb{E}_{X\sim p_X}[\phi(f(X))]\}\le\min_{f\in supp(p_F)}\{\mathbb{E}_{X\sim p_X}[\phi(f(X))]\}$$
> As a result, Theorem 4.1 holds for both minimization regions.
>
> - The distributional equivalence property: The requirement that the reviewer mentioned is a weaker form of the *distributional equivalence* property revealed in our work, whose detailed descriptions and verifications can be found in sec 4.1 of our manuscript. We clarify that it should be "$\hat{F},\hat{G}\sim \hat{p}_F$" instead of for all $\hat{f},\hat{g}\in\hat{\mathcal{F}}$. We will revise this typo in the proof. However, this typo does not affect any of the theoretical results.
>
>  We emphasize that this property is the main cause of the performance gain of homogeneous deep ensembles, which has never been discovered before. Due to the lack of this property, previous work can only refer to ensembles as "a reliable black box" (L61-63).
>
>
> **References**
>
> [1] Taiga Abe, Estefany Kelly Buchanan, Geoff Pleiss, Richard Zemel, and John P Cunningham. Deep ensembles work, but are they necessary? Advances in Neural Information Processing Systems, 35: 33646–33660, 2022b.
>
> [2] Preetum Nakkiran, Gal Kaplun, Yamini Bansal, Tristan Yang, Boaz Barak, and Ilya Sutskever. Deep double descent: Where bigger models and more data hurt. Journal of Statistical Mechanics: Theory and Experiment, 2021(12):124003, 2021.
>
> [3] Kobayashi, Seijin, Johannes von Oswald, and Benjamin F. Grewe. "On the reversed bias-variance tradeoff in deep ensembles."

---

> > ### Comment · Reviewer_wGn1 · 2024-11-26
> > **Response to the rebuttal**
> >
> > Thanks for responding my comments and questions. Indeed, I spent my time to reread this paper deeply including all proof.
> >
> > I appreciate the authors to conduct experiments on many trained models to demonstrate the distributional equivalence property which is mysterious to me. However, I concern about the theoretical set-up of this paper as well as the significance of the results.
> >
> > First, there is no rigorous definition of the set of trained models $F$ and the distribution $p_F$. Specifically, up to what level we should train a model to conclude that it belongs to $F$ or sampled from $p_F$. This is vague to me. It would be clearer if the authors define a trained model as a model that can really minimize the empirical loss for instance.
> >
> > Second, in the proof of Proposition 3.1, the derivation $\min_{f \in F} \phi(f(x)) = \phi(\min_{f \in F} f(x))$ does not hold for monotonically decreasing function.  So the statement is only applicable to monotonically increasing function.
> >
> > Third, regarding to  the distributional equivalence of $\hat{p}(F)$, I believe that this cannot be directly inherited from the one of $p_F$ without rigorous proof. Moreover, the distributional equivalence of $\hat{p}(F)$ implies that if $f,g \sim p_F$, they must have the same accuracy over the general data/label distribution. It raises the question between the well-trained models and their accuracies. As far as I understand, Figures 2,8, and 9 are relevant to $f \sim p_F$ not $\hat{f} \sim p_{\hat{F}}$.
> >
> > Fourth, to me the assumption of the predicted probability collapse to one-hot vector is too strong. At least, the authors need to conduct experiments on their well-trained models to verify this. Certainly, one can consider $\hat{f} \in p_{\hat{F}}$ as a simplified version of $f \sim p_F$, however, Theorem 4.1 seems to be trivial because $E_x[\phi(\hat{f}(x)]$ is constant w.r.t. $\hat{f} \sim p_{\hat{f}}$ because it is $acc(\hat{f})\phi(1) + (1-acc(\hat{f}))\phi(0)$. I also concern about the meaning of the left-side expression $E_x[\phi(E_{F \sim p_{\hat{F}}}[F(x)])]$ because $E_{F \sim p_{\hat{F}}}[F(x)]$ means the accuracy of the family in $p_{\hat{F}}$ when evaluated a specific $x$, so what is the meaning of applying $\phi$ to this accuracy? For ensemble learning, we apply $\phi$ to the prediction of the ensemble model which seems to make more sense to me.
> >
> > Sixth, I appreciate Theorem 4.3 because it explicitly specializes the diversity of component models when predicting $x$, i.e., reducing the set of $x$ that two models  are predicted correctly.
> >
> > Last, in Line 152, the objective function $min_f E_{(x,y) \in {X_{train} \times Y_{train}}}[...]$ is weird to me.

---

> > > ### Author Response · Authors · 2024-12-01
> > > **Follow-up Response to Reviewer wGn1 [1/3]**
> > >
> > > We greatly appreciate the reviewer's effort in reviewing our work and the valuable and detailed comments. Now we are more clear about the reviewer's concerns and can address them more accurately.
> > >
> > > 1. **The definition of trained models $F\sim p_F$.**
> > >
> > > We clarify that this paper focuses on models that interpolate the training set. That is, the set $\mathcal{F} = \\{f:\mathbb{E}\_{\mathcal{X}\_{train}\times\mathcal{Y}\_{train}}[L(f(x),y)]\le\delta\\}$ for a sufficiently small $\delta$. This can be understood as an accuracy of ~100% or almost zero training loss, depending on the choice of $L$. For discriminative models, they hold simultaneously in general. Such a process follows the idea of neural collapse and deep double descent. The specific criteria follow the important work in [1], which provides a practical and consistent standard for training interpolation models. We will clearly define the set of "trained models" $\mathcal{F}$ in the final version as suggested by the reviewer.
> > >
> > > As for the distribution $p_F$ and the models as the random variable $F$, we consider the homogeneous ensemble [5,6], where all ensemble members are trained using identical stochastic algorithms. Thus the source of the stochasticity is purely from the algorithm, which results in a distribution $p_F$ of models in $\mathcal{F}$. In the main manuscript, we follow previous works and use a plain SGD solver. Furthermore, in the updated manuscript, we already included SGD with momentums and weight decays as a complement. All the observations still hold. We will include more variants in the final version for a comprehensive perspective.
> > >
> > > 2. **Proof of Proposition 3.1**
> > >
> > > We would like to point out that proposition 3.1 holds for both increasing and decreasing $\phi$. The provided proof in the rebuttal assumes the metric function $\phi$ to be monotonically increasing without loss of generality. This is because **the proof generalizes to the decreasing $\phi$ case symmetrically.** We elaborate on this as follows:
> > >
> > > Let $\phi$ be monotonically $\color{red}{decreasing}$. Then
> > > $$\min_{f\in\mathcal{F}}\{\phi(f(\mathbf{x}))\}=\phi({\color{red}{\max}}\_{f\in\mathcal{F}}\{f(\mathbf{x})\})$$
> > > Since ${\color{red}{\max}}\_{f\in\mathcal{F}}\{f(\mathbf{x})\}{\color{red}{\ge}}\mathbb{E}\_{F\sim p_F}[F(\mathbf{x})]$ and $\phi$ is monotonically $\color{red}{decreasing}$, we have
> > > $$\phi({\color{red}{\max}}\_{f\in\mathcal{F}}\{f(\mathbf{x})\})\le\phi(\mathbb{E}\_{F\sim p_F}[F(\mathbf{x})])$$
> > > Combining them results in
> > > $$\min_{f\in\mathcal{F}}\{\phi(f(\mathbf{x}))\}\le\phi(\mathbb{E}\_{F\sim p_F}[F(\mathbf{x})])$$
> > > Therefore
> > > $$LG(\mathbf{x})=\min_{f\in\mathcal{F}}\{\phi(f(\mathbf{x}))\}-\phi(\mathbb{E}\_{F\sim p_F}[F(\mathbf{x})])\le0$$
> > > This proves the proposition.$\Box$
> > >
> > > To highlight how they are symmetric, we note the symmetric operations in $\color{red}{red}$. Due to this symmetry, we simplified the proof scenario without loss of generality. We will update the proof in the final version to avoid potential misunderstandings.
> > >
> > > 3. **The distributional equivalence of $\hat{f}$**
> > >
> > >  i. The distributional equivalence of $\hat{p}_F$ (Theoretical)
> > >
> > > First, we show that the distributional equivalence property holds for the surrogate models.
> > > In sec. 4.2, the surrogate models are defined as $\hat{\mathcal{F}}=\\{\hat{f}(x)=\mathbb{1}\_{\arg\max f(x)=y}|f\in\mathcal{F}\\}$. Therefore, $\hat{f}\sim Bernoulli(\rho)$ where $\rho=\mathbb{E}_{X\sim p_X}[\mathbb{1}\_{\\{f\textrm{ correctly predicts }X\\}}]$. Therefore, the distributional equivalence holds for the surrogate model. We will add this clarification to the final version.
> > >
> > >  ii. Same Accuracy (Empirical)
> > >
> > > The reviewer is correct that distributional equivalence automatically leads to **almost identical performance**. That is, when models are drawn from $p_F$ (i.e. trained using the same criterion as described in the answer to point 2), *there testing accuracies are almost identical*. The "identical accuracy" experiments were skipped since "distributional equivalence" is considered a much stronger property. We will add figures showing the identical performance as a complement to the distributional equivalence property in the final version.

---

> > > > ### Author Response · Authors · 2024-12-01
> > > > **Follow-up Response to Reviewer wGn1 [2/3]**
> > > >
> > > > 4. **The neural collapse assumption**
> > > >
> > > > We respectfully disagree with the reviewer that the neural collapse assumption is too strong. It should be noted that this is the existing practice in training deep models as demonstrated in various literatures (e.g. [1-4]).
> > > >
> > > > Besides, as suggested by the reviewer, we have actually carried out numerous experiments to verify the collapse-to-one-hot assumption is valid both theoretically and empirically.
> > > >
> > > > - In Figure 2,8,9, we demonstrate the PDFs of the prediction, which fall extremely closely to $\\{0,1\\}\subset[0,1]$. This empirically motivates the approximation using one-hot predictions.
> > > >
> > > > - We acknowledge the potential compromise and derive a general version of the theoretical results without instantiating any variables in Appendix B.9 (B.8 in the original draft). It shows that "the use of $\hat{p}_F$ is purely to avoid the untractable continuous distributions", which does not sacrifice the insights and contributions of the results.
> > > >
> > > > - This is further verified in **all** experiments, where the theoretical results derived based on $\hat{f}$ accurately describe the behavior of trained models $f$. All experiments are carried out using $f$, and the theoretical results estimate the behavior of $f$ accurately.
> > > >
> > > > As for Theorem 4.1, we emphasize that it is the fact that for trained models $f\in p_F$, their "ensembled predictions" are very close to the "accuracy of the family" when evaluating a specific $x$. This is determined by the neural collapse phenomenon of models. To resolve the reviewer's concern, we inspect the differences between these two values (a) ensembled prediction $e(x)=\mathbb{E}\_{F\sim\hat{p}_F}[F(x)]$ and (b) accuracy of the family $a(x)=\mathbb{E}\_{F\sim p_F}[F(x)]$ by the distribution of $|e(X)-a(X)|$ where $X\sim p_X$. Since the manuscript cannot be updated anymore, here we present the CNN models and all datasets. They are as follows
> > > >
> > > > Tab R1: The mean and standard deviation of $|e(X)-a(X)|$, the difference between the ensembled prediction and the accuracy of the family.
> > > >
> > > > |$k$|10|20|40|80|160|
> > > > |-|-|-|-|-|-|
> > > > |CIFAR-10| 0.019$\pm$0.016|0.021$\pm$0.021|0.024$\pm$0.027|0.024$\pm$0.031|0.022$\pm$0.033|
> > > > |CIFAR-100|0.050$\pm$0.044|0.040$\pm$0.034|0.057$\pm$0.052|0.070$\pm$0.07|0.073$\pm$0.079|
> > > > |TinyImagenet|0.098$\pm$0.106|0.039$\pm$0.04|0.038$\pm$0.037|0.060$\pm$0.062|0.074$\pm$0.081|
> > > >
> > > > It can be observed that the two values are indeed extremely close in practice because of the neural collapse phenomenon. Thus the proof of Theorem 4.1 does not sacrifice any validity. We will add more detailed validations to the final version.
> > > >
> > > > In summary, we emphasize that using succinct modeling to explain a long-lasting mysterious phenomenon should be considered as an advantage instead of a drawback.
> > > >
> > > >
> > > > 5. We did not find the 5th point and deduced that this may be a typo.

---

> > > > > ### Author Response · Authors · 2024-12-01
> > > > > **Follow-up Response to Reviewer wGn1 [3/3]**
> > > > >
> > > > > 6. **Theorem 4.3**
> > > > >
> > > > > We appreciate greatly that the reviewer acknowledges the contribution of our Theorem 4.3, where the performance of ensembles of $M\rightarrow\infty$ models can be estimated by only two models. In fact, we further derive more practical results in Theorem 5.1 to estimate the performance of the ensemble of **arbitrary $M$** models using only two models. And the results are verified in Figure 7.
> > > > >
> > > > > 7. **Notation in L152**
> > > > >
> > > > > We thank the reviewer for pointing out this possible misunderstanding. This is a simplified version of the common notation in optimizing models that are parameterized by $\theta$:
> > > > > $$\min_\theta \mathbb{E}\_{(\mathbf{x},y)\sim\mathcal{X}\times\mathcal{Y}}L(\mathbf{f}_\theta(\mathbf{x}),y)$$
> > > > >
> > > > > Here we simplify this to $$\min_{\mathbf{f}} \mathbb{E}\_{(\mathbf{x},y)\sim\mathcal{X}\times\mathcal{Y}}L(\mathbf{f}(\mathbf{x}),y)$$ to emphasize that these models should be considered as a whole and lead to the analysis in the functional space. We will clarify this in the final version to avoid misunderstanding.
> > > > >
> > > > >
> > > > > **Summary**
> > > > >
> > > > > We would like to thank the reviewer again for the dedicated and meticulous review of our work. We emphasize that we are the first to finally resolve the mystery of deep ensembles. We derive theoretical results that accurately describe the behaviors of ensembles as a "scaling law" and provide valuable insights in designing the ensembles. Besides, the discovered "distributional equivalence" phenomenon is also novel and important.
> > > > > Given the acknowledged contributions by both reviewer wGn1 and other reviewers, we kindly ask that the reviewer reconsider the assessment of our work.
> > > > > And we are more than happy to answer any further questions if the reviewer's concern is not fully resolved.
> > > > >
> > > > >
> > > > > **References**
> > > > >
> > > > > [1] Preetum Nakkiran, Gal Kaplun, Yamini Bansal, Tristan Yang, Boaz Barak, and Ilya Sutskever. Deep double descent: Where bigger models and more data hurt. Journal of Statistical Mechanics: Theory and Experiment, 2021(12):124003, 2021.
> > > > >
> > > > > [2] Vardan Papyan, XY Han, and David L Donoho. Prevalence of neural collapse during the terminal phase of deep learning training. Proceedings of the National Academy of Sciences, 117(40): 24652–24663, 2020
> > > > >
> > > > > [3] Cao, Y., Chen, Z., Belkin, M., and Gu, Q. (2022). Benign overfitting in two-layer convolutional neural networks. Advances in neural information processing systems, 35:25237–25250.
> > > > >
> > > > > [4] Vardan Papyan, XY Han, and David L Donoho. Prevalence of neural collapse during the terminal phase of deep learning training. Proceedings of the National Academy of Sciences, 117(40): 24652–24663, 2020
> > > > >
> > > > > [5] Taiga Abe, Estefany Kelly Buchanan, Geoff Pleiss, Richard Zemel, and John P Cunningham. Deep ensembles work, but are they necessary? Advances in Neural Information Processing Systems, 35: 33646–33660, 2022b.
> > > > >
> > > > > [6] Balaji Lakshminarayanan, Alexander Pritzel, and Charles Blundell. Simple and scalable predictive uncertainty estimation using deep ensembles. Advances in neural information processing systems, 30, 2017

---

> > > > > > ### Comment · Reviewer_wGn1 · 2024-12-02
> > > > > > **Response to the authors**
> > > > > >
> > > > > > Thanks for answering my questions. Although the theories developed in the paper are not absolutely rigorous, its insights are beneficial. I decide to raise my score.

---

### Author Response · Authors · 2024-11-24
**We are happy to answer any questions if there still exist concerns for our paper.**

Dear Reviewers,

Thanks very much for the time and effort that you have dedicated to reviewing our paper. We greatly appreciate your constructive comments and hope our response addresses your concerns adequately.

Should you have any further questions or need further clarification, we are more than willing to provide it. Thank you again for your valuable insights.

Best,

Authors

---

### Meta-Review · Area_Chair_89S5 · 2024-12-20

**Metareview:**

The paper challenges traditional explanations for the effectiveness of deep ensembles and proposes a novel concept: distributional equivalence, where ensemble members' predictions share identical statistical distributions.

Key findings:

- Deep ensembles' effectiveness does not arise solely from Jensen's inequality but from the distributional equivalence property.

- Rigorous theoretical analysis quantifies ensemble improvements, incorporating sample diversity and ensemble size.

- A novel estimation scheme predicts ensemble performance using only two models, offering computational efficiency.

- Empirical results validate the theory across datasets like CIFAR-10 and TinyImageNet, emphasizing the role of distributional equivalence in ensemble performance.

The paper excels by combining novel perspectives, theoretical insights, and practical implications. The reviewers unanimously praised its novelty, practicality, comprehensive evaluation, and clear presentation.

A common concern was the reliance on the neural collapse assumption, which may not hold universally. However, the assumption is applicable to many practical scenarios where neural collapse occurs, making the results in this paper both relevant and impactful. Another potential issue is that the theory is “not absolutely rigorous” (Reviewer wGn1), for which I’d urge the authors to take a closer look to make sure that the paper is clear of technical errors. That said, reviewer wGn1 also recommended acceptance in view of the beneficial insights. Also, upon reading the reviewer's responses to these concerns, it appears more of a clarity issue or a lack of details, which I trust that the authors can address properly. Overall, I find the paper deserving of acceptance.

**Additional Comments On Reviewer Discussion:**

Reviewer wGn1 asked questions on the technical details. After exchanging with the authors for a few rounds, the reviewer concludes that the theories developed in the paper are not absolutely rigorous but thought the  insights are beneficial.

Reviewer fvho had questions on the generalizability of the theory for different loss functions and neural collapse, and thought it might be acceptable although there are still concerns.

Reviewer C32A expressed concern that the theory "significantly limits the generalizability of the findings" due to reliance on the neural collapse assumption. The authors provided additional experiments and clarified that the assumption aligns with standard training practices, but Reviewer C32A indicated that the generalizability concern was not "satisfactorily resolved."

Reviewer CxeF criticized the paper for not discussing the effect of distributional equivalence on uncertainty quantification, a key application area for deep ensembles. This is addressed after the discussion with the authors.

---

### Decision · Program_Chairs · 2025-01-22

Accept (Poster)